# Secondary aerosol formation alters CCN activity in the North China Plain

Jiangchuan Tao[1, 2], Ye Kuang[1, 2], Nan Ma[1, 2], Juan Hong[1, 2], Yele Sun[3, 4, 5], Wanyun Xu[6], Yanyan Zhang[1], Yao He[3], Qingwei Luo[1], Linhong Xie[1, 2], Hang Su[7], Yafang Cheng[7]

[1]Institute for Environmental and Climate Research, Jinan University, Guangzhou, Guangdong 511443, China

[2]Guangdong-Hongkong-Macau Joint Laboratory of Collaborative Innovation for Environmental Quality, Guangzhou, China

[3]State Key Laboratory of Atmospheric Boundary Layer Physics and Atmospheric Chemistry, Institute of Atmospheric Physics, Chinese Academy of Sciences, Beijing 100029, China

[4]College of Earth and Planetary Sciences, University of Chinese Academy of Sciences, Beijing 100049, China

[5]Center for Excellence in Regional Atmospheric Environment, Institute of Urban Environment, Chinese Academy of Sciences, Xiamen 361021, China

[6]State Key Laboratory of Severe Weather, Key Laboratory for Atmospheric Chemistry, Institute of Atmospheric Composition, Chinese Academy of Meteorological Sciences, Beijing, 100081, china

[7]Multiphase Chemistry Department, Max Planck Institute for Chemistry, Mainz 55128, Germany

Correspondence to: Jiangchuan Tao (taojch@jnu.edu.cn) and Nan Ma (nan.ma@jnu.edu.cn)

**Abstract:**

Secondary aerosols (SA, including secondary organic and inorganic aerosols, SOA and SIA) are predominant components of aerosol particles in the North China Plain (NCP) and its formation has significant impacts on the evolution of particle size distribution (PNSD) and hygroscopicity. Previous studies have shown that distinct SA formation mechanisms can dominate under different relative humidity (RH). This would lead to different influences of SA formation on the aerosol hygroscopicity and PNSD under different RH conditions. Based on the measurements of size-resolved particle activation ratio (SPAR), hygroscopicity distribution (GF-PDF), $PM_{2.5}$ chemical composition, PNSD, meteorology and gaseous pollutants in a recent field campaign, McFAN (Multiphase chemistry experiment in Fogs and Aerosols in the North China Plain), conducted during the autumn-winter transition period in 2018 at a polluted rural site in the NCP, the influences of SA formation on cloud condensation nuclei (CCN) activity and CCN number concentration($N_{CCN}$)calculation under different RH conditions were studied. Results suggest that during daytime, SA formation could lead to a significant increase in $N_{CCN}$ and a strong diurnal variation in SPAR at Super-saturations lower than 0.07%. During periods with daytime minimum RH exceeding 50% (high RH conditions), SA formation significantly contributed to the particle mass/size changes in a broad size range of 150 nm to 1000 nm, led to $N_{CCN}$(0.05%) increases within the size range of 200 nm to 300 nm, and mass concentration growth mainly for particles larger than 300 nm. During periods with daytime minimum RH below 30% (low RH conditions), SA formation mainly contributed to the particle mass/size and $N_{CCN}$ changes for particles smaller than 300 nm. As a result, under the same amount of mass increase induced by SA formation, the increase of $N_{CCN}$(0.05%) was stronger under low RH conditions and weaker under high RH conditions. Moreover, the diurnal variations of SPAR parameter (inferred from CCN measurements) due to SA formation varied with RH conditions, which was one of the largest uncertainties within $N_{CCN}$ predictions. After considering the SPAR parameter (estimated through the number fraction of hygroscopic particles or mass fraction of SA), the relative deviation of $N_{CCN}$(0.05%) predictions were reduced to within 30%. This study highlights the impact of SA formation on CCN activity and $N_{CCN}$ calculation, which provides guidance for future improvements of CCN predictions in chemical-transport models and climate models.

## 1. Introduction

Cloud condensation nuclei (CCN) activity of aerosol particles describes its ability to activate and grow into cloud droplets at given supersaturations and thus has important impacts on cloud microphysics and the aerosol indirect effect on climate. CCN activity is dependent on the physicochemical properties of aerosol particles, including particle size distributions, hygroscopicity (determined by chemical composition) and mixing state. Thus, atmospheric processes influencing these aerosol properties may exert influences on CCN activity.

Secondary aerosols (SA) formation contributes greatly to aerosol populations and impacts CCN properties in many ways, generally increasing CCN number concentrations ($N_{CCN}$) and leading to changes in the CCN activity (Wiedensohler et al., 2009; Kerminen et al., 2012; Wu et al., 2015; Farmer et al., 2015; Ma et al., 2016; Zhang et al., 2019 and reference therein). Differences in precursor and oxidant concentrations as well as SA formation mechanisms lead to particle size growth in different size ranges (Dal Maso et al., 2005; Kulmala et al., 2007; Zhang et al., 2012; Farmer et al., 2015; Cheng et al., 2016; Kuang et al., 2020c), thus would impact CCN activities in different ways. SA formation includes both the formation and subsequent growth of new particles (New Particle Formation, NPF), and the growth of existing particles. NPF can directly provide particles large enough to act as CCNs (Wiedensohler et al., 2009; Kerminen et al., 2012; Farmer et al., 2015), generally affecting aerosol particles smaller than 100 nm, thereby elevating $N_{CCN}$ at higher supersaturations (SSs>0.2%) (Wiedensohler et al., 2009; Kerminen et al., 2012; Ma et al., 2016; Zhang et al., 2019 and reference therein). SA formation on existing particles, especially under polluted conditions, significantly adds mass to and changes the chemical composition of accumulation mode particles (Farmer et al., 2015), thus affecting CCN at lower SSs (<0.2%) (Wiedensohler et al., 2009; Mei et al., 2013; Yue et al., 2016; Thalman et al., 2017; Duan et al., 2018). SSs varies greatly among different clouds categories. Cumulus clouds are formed under higher SSs and are thus mostly influenced by Aitken mode particles formed in NPF events (Reuter et al., 2009; Gryspeerdt and Stier, 2012; Fan et al., 2016; Jia et al., 2019 and reference therein). Stratus clouds and fogs that exert stronger effects on climate and environment, however, are generally formed at SSs lower than 0.2%, indicating that only accumulation mode particles can serve as CCN (Ditas et al., 2012; Hammer et al., 2014a, b; Krüger et al., 2014; Shen et al., 2018). Numerous studies have investigated the impact of NPF on CCN (Gorden et al., 2016; Ma et al., 2016; Yu et al., 2020 and reference therein), however, only few studies have focused on the influence of SA formation on CCN activity of accumulation mode particles, which might exhibit strong climate and environment impacts and urgently requires attention.

SA formation affects CCN activity of accumulation mode particles not only by enlarging their
size, but also by changing their chemical compositions. At a specific particle size, the CCN activity is
determined both by the chemical composition of particles, which originally were and stayed this size,
and  that of particles, which grew into this size via added SA mass. These two groups of particles can
exert different variations to CCN activity at the same particle size (Wiedensohler et al., 2009 and
reference therein). In general, the SA formation can increases the hygroscopicity of particles by adding
chemical compounds with lower volatility and higher oxidation state, which are usually more
hydrophilic, thereby enhancing CCN activity of accumulation mode particles (Mei et al., 2013; Yue et
al., 2016). However, CCN activity may also remain unchanged (Wiedensohler et al., 2009) or be
weakened in some cases (Thalman et al., 2017; Duan et al., 2018). In SA formation observed in central
Amazon forests, Thalman et al. (2017) reported enhanced CCN activity in dry season while constant
CCN activity in wet season. In SA formation events under polluted conditions in Guangzhou (Pearl
River Delta, China), Duan et al. (2018) found that bulk CCN activity can be enhanced in summer due
to the formation of large and inorganic-rich particles, but weakened in winter due to the formation of
small and organic-rich particles, where RH seemed to have been an important factor in the variations
of bulk CCN activity due to different particle formation pathways. Aside from variations of particle
chemical composition, changes in aerosol mixing states caused by SA formation can also change CCN
activity (Su et al., 2010; Rose et al., 2011; Cheng et al., 2012). The fast condensation of SA components
on accumulation mode particles led to the turnover of soot particle mixing state from externally to
internally mixed, which contributed mostly to enhancements of CCN activity (Cheng et al., 2012).
Thus, SA formation influences the CCN activity of accumulation mode particles through its integrated
impacts on their size, hygroscopicity and mixing state, which requires more detailed and
comprehensive investigations.
The North China Plain (NCP) frequently experiences severe aerosol pollution due to both
strong emissions of primary aerosol and strong SA formation caused by the abundance of gaseous
precursors and oxidants (Zheng et al. ACP, 2015; Liu et al., 2010; Huang et al., 2014; Xu et al., 2019).
In the SA formation events on the NCP, both aqueous-phase processes and gas-phase photochemical
processes can play important roles, depending on atmospheric conditions such as RH (Hu et al., 2016;
Xu et al., 2017a; Wang et al., 2019). A recent observational study on the NCP found that SA formation
dominantly contributed to different particle size since SA formation mechanisms varied with RH
conditions (Kuang et al., 2020c). Under dry conditions, SA were mainly formed through gas-phase
photochemical processing and mostly added mass to accumulation mode particles. While under high
RH conditions or super-saturated conditions, SA was also formed in aqueous phase,  contributing to
the formation of both accumulation mode and coarse mode particles. The difference in particle size
where SA formation took place and the difference in SA chemical compositions could result in distinct
variations of CCN activity, which has not been evaluated yet. In this study, we will study the influence
of SA formation on Size-resolved Particle Activation Ratio (SPAR) of accumulation mode particles
in the NCP under different RH conditions, which fills a gap of knowledge within CCN studies in the
NCP and may provide guidance for the improvement of current CCN parameterization schemes in
chemical-transport and climate models.
**2. Method:**
2.1. Measurements
2.1.1. Site
Under the framework of McFAN (Multiphase chemistry experiment in Fogs and Aerosols in
the North China Plain) (Li et al., 2021), from 16$^{th}$ November to 16$^{th}$ December 2018, physical and
chemical properties of ambient aerosol particles as well as meteorological parameters were
continuously measured at the Gucheng site in Dingxing county, Hebei province, China. This site is an
Ecological and Agricultural Meteorology Station (39°09'N, 115°44'E) of the Chinese Academy of
Meteorological Sciences, which is located between Beijing (~100km) and Baoding (~40km), two
mega cities in the North China Plain, and surrounded by farmlands and small towns. Measurements at
this site can well represent the polluted background conditions of the NCP. All aerosol measurement
instruments were placed in a container with temperature maintained at 24 °C, while conventional trace
gas instruments including CO were housed in an air-conditioned room on a two-story building located
~80 meters to the south of the container, with no taller buildings between them blocking the air flow.
2.1.2 Instrumentation
In this study, ambient aerosol was sampled by an inlet system consisting of a PM10 inlet
(Rupprecht & Patashnick Co., Inc., Thermo, 16.67 L/min), a Nafion dryer that dried relative humidity
to below 30% and an isokinetic flow splitter directing the air sample to each instrument.
A DMA-CCNC system measured SPAR at five supersaturations (SSs), 0.05%, 0.07%, 0.2%,
0.44% and 0.81%, with a running time of 20 min for 0.05% and 10 min for the other SSs. This system
consisted of a differential mobility analyzer (DMA model 3081; TSI, Inc, MN USA), a condensation
particle counter (CPC model 3772; TSI, Inc., MN USA) and a continuous-flow CCN counter (model
CCN200, Droplet Measurement Technologies, USA; Roberts and Nenes, 2005). The system was
operated in a size-scanning mode over the particle size range from 9 to 400 nm. SPAR can be obtained
by combining the measurements of CPC and CCNC at different particle size. The sample and sheath
flow rate of the DMA were set to 1 lpm and 5 lpm, respectively, hence the resultant measured particle
diameter ranged from 9 nm to 500 nm. Since the low number concentration of particles above 300 nm
could lead to large uncertainty in CCNC counting, the measurements for particles larger than 300 nm
were excluded, except for 0.05% SS. In order to characterize the variations of particles with low
hygroscopicity of about 0.1, SPAR measurement up to about 400 nm is used at 0.05% SS. There are
12 size distribution scans during a complete 1-hour cycle, with four scans for first SS and two scans
for each of the rest four SSs. Only the last scan for each SS is used as the CCNC needs time for SS
stabilization. The SSs of CCNC were calibrated with monodispersed ammonium sulphate particles
(Rose et al., 2008) both before and after the campaign. The flowrates were checked regularly (every
few days) during the campaign, as the flows (sample flow and sheath flow) of the instrument can affect
both the counting of droplets and the SS in the column. A modified algorithm based on Hagen and
Alofs (1983) and Deng et al. (2011, 2013) was used to correct the influence of multiple-charge particles
and DMA transfer function on SPAR. Details about the system are described in Ma et al. (2016) and
the description about the inversion method can be found in the supplements.
Non-refractory particulate matter (NR-PM) including $SO_4^{2-}$, $NO_3^-$, $NH_4^+$, $Cl^-$ and organics with
dry aerodynamic diameters below 2.5 μm was measured by an Aerodyne Time-of-Flight Aerosol
Chemical Speciation Monitor (ToF-ACSM hereafter) equipped with a $PM_{2.5}$ aerodynamic lens
(Williams et al., 2010) and a capture vaporizer (Xu et al., 2017b; Hu et al., 2017a) at 2-minute time
resolution. The ToF-ACSM data were analyzed with the standard data analysis software (Tofware
v2.5.13; https://sites.google.com/site/ariacsm/, last access: 21 January 2020). The organic mass spectra
from *m/z* 12 to 214 were analyzed with an Igor Pro based positive matrix factorization (PMF)
evaluation tool (v3.04) and then evaluated following the procedures described in Zhang et al. (2011).
The chosen five-factor solution includes four primary factors i.e. hydrocarbon-like OA (HOA),
cooking OA (COA), biomass burning OA (BBOA), and coal combustion OA (CCOA), and a
secondary factor, i.e. oxygenated OA (OOA). More detailed descriptions on the ACSM measurements
and data analysis can be found in Kuang et al. (2020b) and Sun et al. (2020).
A Humidified Tandem differential mobility analyzer (HTDMA, Tan et al., 2013) measured the
size-resolved aerosol growth factor (GF) at 90% RH. The sampled particles were subsequently charged
by a neutralizer (Kr85, TSI Inc.) and size selected by a DMA (DMA1, model 3081L, TSI Inc.). A
Nafion humidifier (model PD-70T-24ss, Perma Pure Inc., USA) was used to humidify the
monodisperse particles with a specific diameter ($D_d$) at a fixed RH of (90 ± 0.44) % and then the
number size distribution of the humidified particles ($D_{wet}$) was measured by another DMA (DMA2,

model 3081L, TSI Inc.) and a condensation particle counter (CPC, model 3772, TSI Inc.). Thus, GF of the particles can be calculated as:

$$GF = \frac{D_{wet}}{D_d} \qquad (1)$$

During the campaign, four dry mobility diameters (60, 100, 150, and 200 nm) were selected for the HTDMA measurements. A full scan takes about 1 hour in order to cover the four sizes. Regular calibration by using standard polystyrene latex spheres and ammonium sulfate were performed to ensure the instrument functioned normally. The tandem differential mobility analyzer (TDMA) inversion algorithm (Gysel et al., 2009) was applied to calculate the Probability Density Function of GF (GF-PDF). More details about this system can be found in Cai et al. (2018) and Hong et al. (2018).

Particle number size distributions (PNSDs) were measured by combining the measurements of a scanning mobility particle sizer (SMPS, TSI model 3080) and an aerodynamic particle sizer (APS, TSI Inc., Model 3321), that measured particle mobility diameter size distributions in the range of 12 nm to 760 nm and particle aerodynamic diameter size distribution in the range of 700 nm to 10 μm, respectively. A commercial instrument from Thermo Electronics (Model 48C) was used to measure CO concentration. Besides monthly multipoint calibrations and weekly zero-span check, additional 6-hourly zero checks were also performed for the CO instrument.

## 2.2. Data processing

### 2.2.1. Aerosol hygroscopicity and cloud activation: κ-Köhler theory

The ability of particles to act as CCN and its dependence on particle size and particle chemical composition on CCN activity can be described by the Köhler theory (Köhler, 1936). A hygroscopic parameter κ is calculated based on the κ-Köhler theory (Petters and Kreidenweis. 2007) to evaluate the influence of particle chemical compositions:

$$\kappa = \left(\frac{D_{wet}^3 - D_d^3}{D_d^3}\right)\left[\frac{1}{S}\exp\left(\frac{4\sigma_{s/a}M_w}{RT\rho_w D_{wet}}\right) - 1\right], \qquad (1)$$

where $S$ represents the saturation ratio, $\rho_w$ is the density of water, $M_w$ is the molecular weight of water, $\sigma_{s/a}$ is the surface tension of the solution/air interface, $R$ is the universal gas constant, T is the temperature, $D_d$ is the diameter of dry particle and $D_{wet}$ is the diameter of the humidified particle. In this study, $\sigma_{s/a}$ is assumed to be the surface tension of pure water/air interface. Based on the κ-Köhler theory, the surface equilibrium water vapor saturation ratio of particles with a specific κ at different wet particle size can be calculated, and the maximum value of surface equilibrium saturation ratio (which is generally supersaturated) is defined as the critical SS for CCN activation. As a result, the

variation of the critical diameter ($D_a$) for particles with different hygroscopicity (or GF at a specific
RH) at different SSs can be determined.

2.2.2. Aerosol growth factor and its probability density function

In practice, the growth factor probability density function (GF-PDF) was inversed from the
measured GF distribution using a TDMAinv algorithm (Gysel et al., 2009). After obtaining the GF-
PDF, the ensemble average GF and corresponding critical diameter under a certain SS ($D_{a,GF}$) can be
calculated. Furthermore, the number fraction and the weighted-average GF of hygroscopic particles
($\kappa >0.1$ and GF(90%, 200 nm)>1.22) were calculated as:

$$NF_{hygro}= \int_{1.2}^{\infty} PDF(GF) \times dGF \qquad (2)$$

$$GF_{hygro}= \int_{1.2}^{\infty} GF \times PDF(GF) \times dGF \qquad (3)$$

Based on the $\kappa$-Köhler theory, the hygroscopicity parameter $\kappa$ and corresponding critical diameter
($D_{a,hygro}$) under a certain SS for particles with $GF_{hygro}$ can be calculated. As $GF_{hygro}$ is higher than the
average GF, $D_{a,hygro}$ is smaller than $D_{a,GF}$.

2.2.3 Calculations of aerosol hygroscopicity from aerosol chemical-composition measurements

For the calculation of aerosol hygroscopicity parameter $\kappa$ based on measured chemical
composition data ($\kappa_{chem}$), detailed information on the chemical species are needed. The ACSM can
only provide bulk mass concentrations of $SO_4^{2-}$, $NO_3^-$, $NH_4^+$, $Cl^-$ ions and organic components, which
cannot be used to calculate size resolved hygroscopicity. However, in the North China Plain,
accumulation mode particles are the dominant contributors to the bulk particle mass concentration (Liu
et al., 2014; Xu et al., 2015; Hu et al., 2017b) and thus the bulk chemical compositions can be used as
a proxy for that of accumulation mode particles. For the inorganic ions, a simplified ion pairing scheme
was used to convert ion mass concentrations to mass concentrations of corresponding inorganic salts
(Gysel et al., 2007; Wu et al., 2016). Thus, mass concentrations of $SO_4^{2-}$, $NO_3^-$, $NH_4^+$ and $Cl^-$ are
specified into ammonium sulfate (AS), ammonium nitrate (AN), ammonium chloride (AC) and
ammonium bisulfate (ABS), for which the $\kappa$ values under super-saturated conditions were specified
according to Petters and Kreidenweis (2007). For a given internal mixture of different aerosol chemical
species, the Zdanovskii–Stokes–Robinson (ZSR) mixing rule can be applied to predict the overall $\kappa_{chem}$
using volume fractions of each chemical species ($\varepsilon_i$) (Petters and Kreidenweis, 2007):

$$\kappa_{chem}= \sum_i \kappa_i \cdot \varepsilon_i \qquad (4)$$

where $\kappa_i$ and $\varepsilon_i$ represent the hygroscopicity parameter $\kappa$ and volume fraction of chemical component
$i$ in the mixture. Based on Eq.2, $\kappa_{chem}$ can be calculated as follows:
$$\kappa_{chem}=\kappa_{AS}\varepsilon_{AS}+\kappa_{AN}\varepsilon_{AN}+\kappa_{ABS}\varepsilon_{ABS}+\kappa_{AC}\varepsilon_{AC}+\kappa_{BC}\varepsilon_{BC}+\kappa_{Org}\varepsilon_{Org} \qquad (5)$$
where $\kappa_{BC}$ is assumed to be zero as black carbon is non-hygroscopic. $\kappa_{org}$ and $\varepsilon_{org}$ represent $\kappa$ and
volume fraction of total organics. The values of hygroscopicity parameter for inorganic compounds
can be found in Table 1 of Kuang et al. (2020b). Large variations in $\kappa_{org}$ has been reported in former
studies and a linear relationship between $\kappa_{org}$ and organic aerosol oxidation state (f44) was detected in
our campaign (Kuang et al., 2020b), which was adopted to calculate $\kappa_{org}$ in this study:
$$\kappa_{Org}=1.04\times f44-0.02 \qquad (6)$$
It should be noted that the $\kappa$-Köhler theory is not perfect, even for inorganic compounds.
Numerous studies have been focusing on the performance of its application on measurements under
different RH conditions (Liu et al., 2011; Wang et al., 2017). And $\kappa_{org}$ used in this study was
determined by the measurement of humidified nephelometer at RH of 85% in Petters and Kreidenweis,
(2007), due to the lack of $\kappa_{org}$ measured under super-saturated conditions. In this study, we focus on
the variations of $\kappa$ values derived from HTDMA and CCN measurement during the SA formation
events, rather than the closure between $\kappa$ values derived using different techniques, which will be
addressed in an upcoming study.
2.2.4. Fitting parameterization scheme of SPAR
In general, the variation in CCN activity of a particle population can be attributed to the
variation in the number fraction of hygroscopic particles or its hygroscopicity, which can be indicated
by fitting parameters of SPAR curves parameterization. SPAR curves are often parameterized using a
sigmoidal function with three parameters. This parameterization assumes aerosols to be an external
mixture of apparently hygroscopic particles that can act as CCN and non-hygroscopic particles that
cannot be measured by CCNC within the measured particle size range below 400 nm (Rose et al.,
2010). SPAR ($Ra(D_p)$) at a specific SS can be described as follows (Rose et al., 2008):
$$Ra(D_p)=\frac{MAF}{2}\left(1+erf\left(\frac{D_p-D_a}{\sqrt{2\pi\sigma}}\right)\right) \qquad (7)$$
where erf is the error function. MAF is the asymptote of the measured SPAR curve at large particle
sizes. $D_a$ is the midpoint activation diameter and is associated with the hygroscopicity of CCNs. $\sigma$ is
the standard deviation of the cumulative Gaussian distribution function and indicates the heterogeneity
of CCN hygroscopicity. As reported by Jiang et al. (2021), based on the investigation of the

covariations between SPAR curves and parameterized hygroscopicity distribution, it was found that the MAF can be used to estimate the number fraction of hygroscopic (thus CCN-active) particles, for aerosol hygroscopicity distributions generally observed in ambient atmosphere, and thus half MAF can be used to represent the number fraction of CCNs to total particles at particle size around $D_a$. Although the influence of particles whose $\kappa$ is less than 0.1 on SPAR cannot be considered in this parameterization scheme, significant deviation were only found under higher SSs (Tao et al., 2020) and need not to be considered under the low SSs discussed in this study.

To be noted, the meaning of MAF can be different regard to the SS, and SPAR measurement up to about 400 nm is needed for the MAF fitting for SPAR at SS of 0.05% to represent the particles with $\kappa$ value higher than 0.1. For SPAR at SS of 0.8%, MAF should be 1 at 400 nm diameter. However, a MAF of 1 in this case can lead to overestimations of hygroscopic particle number fraction due to the significant difference between SPAR curves and sigmodal fitting curves. In the former study on SPAR fitting curves in the NCP, it was found that a fitting parameterization with the combination of two sigmodal fitting curves was needed for SPAR fitting at SSs higher than 0.4% (Tao et al., 2020). However, in this study, we investigate SA formation on accumulation mode particles and particle CCN activity at SSs below 0.1%, under which condition non-hygroscopic particles smaller than 400 nm are typically CCN-inactive. The MAF fitted in the particle size range below 400 nm was used to indicate the variations of SPAR that was of the main focus here in this work. In addition, due to the very low $N_{CCN}$ in particle size ranges larger than 400 nm, the deviations of $N_{CCN}$ due to the limited range of measured particle size are also very small.

## 3. Results

### 3.1. Overview of the measurements

The timeseries of meteorological parameters, SPAR, $N_{CCN}$ at SS of 0.05% and mass concentration of Non-refractory particulate matter of PM$_{2.5}$ (NR-PM$_{2.5}$), PM$_{2.5}$ SA (inorganic compounds and OOA) and PM$_{2.5}$ PA (primary aerosol, defined as the sum of POA) are shown in Fig. 1. The mass concentration of OOA and four POA were quantified by the ACSM PMF analysis (Zhang et al. 2011). During the campaign, PM$_{2.5}$ PA were generally lower than 100 μg m$^{-3}$ under both high and low RH periods. Meanwhile, PM$_{2.5}$ SA can approach about 400 μg m$^{-3}$, especially during the strong SA formation events under high RH conditions, but can be lower than 100 μg m$^{-3}$ under low RH conditions. Strong diurnal variations were found in SPAR with varying meteorological parameters. During the whole period, the wind speed was generally lower than 4 m s$^{-1}$, which is in favor of aerosol particle accumulation and SA formation on existing particles. However, RH, $N_{CCN}$(0.05%), PM$_{2.5}$ SA and NR-PM$_{2.5}$ mass concentrations revealed very distinct levels before and after 4$^{th}$ Dec, and thus the whole campaign was divided into two stages with different RH and SA pollution conditions: higher RH and stronger SA pollution before 4$^{th}$ Dec, and lower RH and lighter SA pollution after 4$^{th}$ Dec. In the following discussions, the high RH stage corresponds to days before 4$^{th}$ Dec with daily maximum and minimum RH higher than 75% and 50%, respectively. Two events that occurred during 25$^{th}$ Nov to 27$^{th}$ Nov (Event 1) and 30$^{th}$ Nov to 2$^{nd}$ Dec (Event 2), respectively, displayed especially high RH conditions with successive nighttime fogs (blue shaded areas). The low RH stage corresponds to the period after 4$^{th}$ Dec with daily maximum and minimum RH below 70% and 30%, which was represented by two events that occurred during 9$^{th}$ Dec to 11$^{th}$ Dec (Event 3) and 13$^{th}$ Dec to 15$^{th}$ Dec (Event 4), respectively. These events were selected based on the similarity of PM$_{2.5}$ concentration and evolution, while the time window was fixed to two days for the convenience of intercomparing. In addition, during these events, the wind speed was generally low, the RH followed a general diurnal variations and SA mass grew steadily and continuously. Thus the interference of the variations of air mass and short-term local emissions can be eliminated and the influence of SA formation can be highlighted. It should be noted that variations of $N_{CCN}$ at 0.07% were similar to those at 0.05%, which followed the variations of SA mass concentration. While at higher SSs, the variations of $N_{CCN}$ differed from those of SA mass concentration, especially under high RH conditions, suggesting different responses of CCN activity towards distinct SA formation processes. As reported in Kuang et al. (2020c), during the high RH stage aqueous phase SA formation was promoted, leading to persistent increases in $N_{CCN}$(0.05% and 0.07%), mass concentration of NR-PM$_{2.5}$ and especially mass concentration of PM$_{2.5}$ SA during Event 1 and 2. During the low RH stage, the SA formation

dominantly occurred in the gas-phase, that generated much less SA than aqueous-phase formation
(Kuang et al., 2020c). Thus, the persistent increases of $N_{CCN}$(0.05% and 0.07%) and $PM_{2.5}$ during
Event 3 and 4 was much weaker than those in Events 1 and 2. Due to the different SA mass fractions,
SPAR during the high RH stage was generally higher than that during the low RH stage. However, the
ratios between $N_{CCN}$(0.05%) and mass concentration of $PM_{2.5}$ SA or NR-$PM_{2.5}$, were lower during the
high RH period and demonstrated strong decreases, especially in Event 1 and 2. The response of CCN
activity and $N_{CCN}$(0.05%) to the different SA formation mechanisms will be discussed
comprehensively in the following parts.

**3.2. The influence of different secondary aerosol formation on the diurnal variation of CCN activity**

The diurnal averages of PNSD, SPAR at SS of 0.05%, GF-PDF for 200 nm particle and mass
fraction of particle chemical compositions during high RH periods before 4[th] Dec, low RH periods
after 4[th] Dec and the four events are shown in Fig. 2, respectively. To be noted, the "high (or low) RH
events" is used to refer to the SA formation events under high (or low) RH conditions for convivence,
and it doesn't mean that RH caused variations of CCN behavior. As can be seen in Figs. 2 (1b) and
(2b), different variations of SPAR due to SA formations can be found during the periods with different
RH conditions. The average diurnal variations of these parameters for the entire high RH stage and
low RH stage as shown in Figs. 2 (1a-1d) and (2a-2d) revealed similar but more smoothed variations
as in the four selected events. The four events are discussed and intercompared in the following to
magnify the differences under distinct RH conditions. For accumulation mode particles, particle
number concentrations were higher during daytime in high RH events, while stronger diurnal
variations occurred in low RH events. Simultaneous daytime increases in particle SPAR in size range
from 200 nm to 400 nm, GF-PDF in GF range from 1.2 to 1.8 and SA mass fraction were found in all
four events, suggesting that SA formation led to increasing hygroscopic particles number
concentration, which in turn enhanced particle CCN activity. This effect was more pronounced in
Events 1 and 2 than in Events 3 and 4. In Events 1 and 2, SPAR values were generally higher than 0.4
at 200 nm and reached the maximum of 1 during noontime at 300 nm. A hygroscopic mode with
GF>1.4 was found throughout the day, which dominated aerosol hygroscopicity during daytime. Mass
fraction of SA were generally higher than 70% and reach a maximum of 80% at noon. While in Events
3 and 4, SPAR at 200 nm was lower than 0.4 at night and the maximum SPAR at 300 nm was lower
than 0.9. A particle mode with GF<1.1 dominates particle hygroscopicity, and the mass fraction of SA
was lower than 60% and 30% at noon and at night, respectively. However, stronger daytime increase
of SA mass fraction and accordingly larger variation in SPAR was observed in Events 3 and 4 than in
Events 1 and 2.

Besides SS of 0.05%, variations of SPAR at SSs of 0.07% and 0.2% are also shown in Figs.

S1 and S2 in the supplement. And as shown in Figs. S1 and S2, the variations of SPAR and $N_{CCN}$/PM
at SS of 0.07% are similar but lighter, compared with those at SS of 0.05%. While for SS of 0.2%, the
difference of SPAR between different periods or events are smaller (Fig. S1), and so did the diurnal
variations of SPAR and GF-PDF at particle size of 100 nm (Fig. S2). Because CCN activity at SS of
0.2% was strong enough (indicated by SPAR value close to 1) in particle size range where the SA
formation dominates, and thus the different SA formations under high or low RH conditions cannot
lead to significant variations of CCN activity at SS of 0.2%. In summary, based on CCN measurements
in this study, the RH-dependent influence of SA formation on CCN activity can be found obviously at
SSs of 0.05% and 0.07%. As the variations of SPAR at SS of 0.07% were quite similar to those at SS
of 0.05%, further analysis was only based on CCN activity at SS of 0.05%.

In Fig. 3a, detailed comparison of particle CCN activity during SA formation events of $N_{CCN}$

enhancements at SS of 0.05% under different RH conditions are shown as the variations of SPAR
curves. Particle CCN activity in Events 1 and 2 were combined due to their similar diurnal variations
(as shown in Fig. 2). Besides SPAR curves (Fig. 3a), corresponding fitting parameters of the SPAR
curve including $D_a$ and MAF were also shown in Figs. 3b and c, respectively, as enhanced SPAR for
particle population can be attributed to hygroscopic particle number fraction increase (MAF increase)
or enhancement of hygroscopic particle hygroscopicity ($D_a$ decrease). Same as demonstrated in Fig. 2,
SPAR was generally higher and thus particle CCN activity(0.05%) were generally stronger in high RH
events than those in low RH events. However, as shown in Fig. 3a, the difference between SPAR in
high and low RH events at 300 nm decreased from 0.2 to 0.1 during the SA formations, indicating for
a stronger enhancement in low RH events, probably due to both the stronger increase of SA mass
fraction and the higher nighttime PA mass fraction (Fig. 2(e)). Furthermore, in high RH events, there
were daytime enhancements of SPAR within the 150 to 300 nm size range, as was indicated by the
daytime increase of MAF and decrease of Da, which mainly resulted from number fraction and
hygroscopicity increases of CCN-active particles. While in low RH events, the daytime enhancement
of SPAR was only observed for particles larger than 200 nm. This can be attributed to the strong
increase of MAF and the slight decrease of $D_a$, which indicates significant increasing number fraction
yet slightly enhanced hygroscopicity of hygroscopic particles, respectively. Overall, the enhancement
of SPAR was weaker but occurred at a broader particle size range in high RH events than in low RH
events, as shown in Fig. 3a. This is in accordance with previous the results from Kuang et al. (2020c),
suggesting that SA formation occurred mainly in aqueous phase within a broad particle size range (up
to 1 μm) in high RH events, while SA formation dominantly proceeded via gas phase reactions and
contributed to aerosol sizes smaller than 300 nm in low RH events. At SS of 0.05% (Fig. 3(a)), the
variation of SPAR from 8:00-12:00 to 12:00-16:00 in particle size smaller than 200 nm was very small
during low RH events, suggesting a smaller CCN activity enhancement due to SA formation compared
with those at high RH events. In detail, the different variations of SPAR in high and low RH events
indicated by MAF and $D_a$ shown in Figs. 3(b & c) suggested different variations of hygroscopicity,
number fraction and size of SA particles. Before SA formation, there was a significant difference
between the MAF in high and low RH events, which disappeared after the SA formation. The stronger
variations in MAF in low RH events suggested stronger enhancement of number concentration of
formed SA particles. As for $D_a$ during SA formation, there were similar, little decrease in both high
and low RH events, suggesting similar hygroscopicity of the SA formed under low and high RH
conditions. Thus differences of SPAR and the resultant $N_{CCN}$ during low and high RH events were
mainly due to the different variations of number fraction of formed SA particles.
As there were different influences of SA formation on both CCN activity at SS of 0.05% and
PNSD under different RH conditions, different variation of $N_{CCN}(0.05\%)$ due to SA formation can also
be expected. Fig. 4 displays the diurnal variation of $PM_{2.5}$ mass concentration, volume concentration
(Vconc) , number concentration (Nconc) and $N_{CCN}(0.05\%)$ (all divided by CO to partially compensate
for changes in planetary boundary layer height), as well as the $N_{CCN}/PM_{2.5}$ mass concentration ratio
and SPAR during high and low RH events, respectively. Variables in Fig. 4 were also presented in Fig.
S3 averaged for the entire high RH and low RH stages, respectively. Compared with the selected case
events featuring significant $N_{CCN}$ enhancement (Fig. 4(1c-2c)), the diurnal variations averaged for the
entire high and low RH stages were similar, with higher levels of particle mass concentration but
weaker enhancement of SA and $N_{CCN}$, indicating similar but weakened impact of SA formation on
CCN activity due to the interference of other aerosol processes. Hereinafter, we discuss the variations
in the four events to magnify the discrepancies of SA formation under high RH and low RH conditions
and its distinct impact on $N_{CCN}$. The Vconc size distribution variations can be used as a proxy for the
evolution of NR-$PM_{2.5}$ size distributions, considering the relatively small variations in particle density
(ranging from 1.2 to 1.8 and with relative variations within 20% (Hu et al., 2012; Zhao et al., 2019)).
The variations of the ratio between $N_{CCN}$ (in different particle size range) and the mass concentration
of $PM_{2.5}$ SA (referred as to $N_{CCN}/SA$) or NR-$PM_{2.5}$ (referred as to $N_{CCN}/NR$) can be used to evaluate
the response of $N_{CCN}$ to SA formation.

During high RH events, normalized $N_{CCN}(0.05\%)$ increased by ~ 50% from 8:00 to 14:00, with a similar increase in normalized $PM_{2.5}$ SA mass concentration (Fig. 4(1a)). As the $PM_{2.5}$ PA mass concentration decrease was much smaller than the SA increase, the $NR-PM_{2.5}$ mass concentration increase can be expected to be similar to the SA increase. As reported by Kuang et al. (2020c), SA during daytime were mainly formed at larger particle sizes, featuring Vconc increase in the particle size range of 400 to 1000 nm. In Fig. 4(1d), significant increases of particle number concentration (Nconc) in particle size range of 150 nm to 1000 nm can be observed. At larger particle size the increase of Nconc led to stronger increase of Vconc, which is why there was simultaneous but much weaker increases of Vconc in particle size range of 150 to 300 nm compared with increases of those in particle size of larger than 300 nm (Fig. 4(1b)). This suggests that $PM_{2.5}$ SA mainly contributed to particle sizes of larger than 300 nm. In addition, because the SA formation enhanced hygroscopicity and number fraction of CCN-active particles in particle size range of 150 to 300 nm, simultaneous enhancements of SPAR can be found throughout the measured particle size range of 180 to 300 nm (Fig. 4(1e)). By combining the enhancements of Nconc and SPAR in measured particle size ranges, there were increases of $N_{CCN}$ from 200 to 500 nm (Fig. 4(1c)). Thus while SA formation processes contributed to their volume (mass) and hygroscopicity increase, it had no further impact on $N_{CCN}$. As a result, $N_{CCN}$ (>300 nm)/SA, $N_{CCN}$ (<300 nm)/SA, $N_{CCN}$ (>300 nm)/NR and $N_{CCN}$ (<300 nm)/NR all decreased during the SA formation (Fig. 4(1f)), suggesting that weakening enhancement of $N_{CCN}(0.05\%)$ in SA formation under high RH condition as SA formation mainly added mass to already CCN-active particles .

During low RH events, weaker increases of both $N_{CCN}(0.05\%)$ and $PM_{2.5}$ SA mass concentration from 8:00 to 14:00 was found (Fig. 4(2a)). At the same time, PA mass decreased by 50% and the variation of total NR mass was small. Under low RH conditions, SA formation mainly contributed to mass enhancements of smaller particle sizes (Kuang et al., 2020c). Vconc increased mostly in the range of 150 to 300 nm (Fig. 4(2b)), while Nconc only increased within 300 nm (Fig. 4(2d)), suggesting that $PM_{2.5}$ SA mainly formed in particle size range below 300 nm. SA formation mainly enhanced number fraction of CCN-active particles in particle size of 200 to 300 nm, as SPAR only revealed evident enhancement (Fig. 4(2e)) and $N_{CCN}$ only significantly increased (Fig. 4(2c)) in that size range. As a result, although $N_{CCN}$ (>300 nm)/SA decreased similar as that under high RH conditions, $N_{CCN}$ (<300 nm)/SA and $N_{CCN}$ (>300 nm)/NR generally stayed constant and $N_{CCN}$ (<300 nm)/NR even increased during SA formation in daytime (Fig. 4(2f)). The ratio between bulk $N_{CCN}$ and mass concentration of $NR-PM_{2.5}$ became larger due to the SA formation, suggesting that

stronger enhancement of $N_{CCN}(0.05\%)$ in SA formation under low RH condition, because SA formation mainly added mass to CCN-inactive particles and turned them into CCN-active particles.

In summary, during the campaign in this study, two kinds of SA formation events were observed under different RH conditions with different variations of PM and $N_{CCN}$ at SSs lower than 0.07%. Under high RH conditions, there were strong SIA dominated SA formation leading to stronger enhancements of CCN-active particle number fraction and $N_{CCN}$. Meanwhile, under low RH conditions, there were moderate SOA dominated SA formation with moderate enhancements of CCN-active particle number fraction and $N_{CCN}$. However, for a unit amount of SA formation, the increase of $N_{CCN}$ was stronger under low RH conditions and weaker under high RH conditions. This was because SA formation under low RH conditions was more concentrated on particle sizes smaller than 300 nm and added more mass to CCN-inactive particles, turning them into CCN-active particles. In addition, strong and distinct diurnal variations of CCN activity of particles were observed during different SA formation processes, whose effects on $N_{CCN}$ calculation need to be further evaluated.

## 3.3. The influence of diurnal variation of CCN activity on $N_{CCN}$ prediction

Since PNSD measurements are generally simpler and more common than $N_{CCN}$ measurements, $N_{CCN}$ is usually estimated from real-time PNSD combined with parameterized CCN activity. In former sections, it was already manifested that SA formations under different RH conditions led to distinct variations in PNSD and SPAR at SS of 0.05%, hence different variations in $N_{CCN}$. Thus, it is important for the prediction of $N_{CCN}$ to quantify its sensitivity towards changes in PNSD and SPAR during SA formation processes under different RH conditions.

In this study, $N_{CCN}$ was mostly determined by PNSD, as was generally the case in former studies (Dusek et al., 2006). Suring SA formation events, however, the variation of CCN activity also contributed significantly to the deviation of $N_{CCN}$ calculation. In former discussions, CCN activity (indicated by SPAR) at 0.05% SS revealed significant diurnal variations during this campaign, which were different during SA formations under distinct RH conditions. The ratio of $N_{CCN}$ calculated based on campaign averaged SPAR ($N_{CCN\_cal}$) to those measured at 0.05% SS ($N_{CCN\_meas}$) before and after 4[th] Dec are shown in Fig. 5. SPAR is determined by the variation of $D_a$ and MAF, which reflect changes in hygroscopicity and number fraction of hygroscopic particles. Thus, to investigate the respective influences of MAF and $D_a$ variations on $N_{CCN}$ predictions, $N_{CCN\_AvgMAF}$ (or $N_{CCN\_avgDa}$) was calculated based on the real-time PNSD and SPAR estimated by replacing MAF (or $D_a$) in Eq. 7 with the campaign averaged value. During the high RH stage, underestimation of daytime $N_{CCN\_cal}$ can reach

up to 20%, since SPAR variations due to CCN activity enhancement were not considered. Similar
deviations of both $N_{CCN\_AvgMAF}$ and $N_{CCN\_avgDa}$ from $N_{CCN\_meas}$ were detected, suggesting that both
MAF and $D_a$ variations contributed to $N_{CCN\_cal}$ underestimation under high RH conditions. During the
low RH stage, up to 50% overestimation existed in $N_{CCN\_AvgSPAR}$ outside  SA formation time periods.
Only $N_{CCN\_AvgMAF}$ displayed similar deviations from $N_{CCN\_meas}$ as $N_{CCN\_AvgSPAR}$, indicating that
differences between $N_{CCN\_cal}$ and $N_{CCN\_meas}$ were mainly contributed by variations in MAF brought on
by significant CCN-active particles number fraction growth due to SA formations. To be noted,
$N_{CCN\_AvgSPAR}$ before and after 4[th] Dec were both calculated based on the SPAR averaged over the entire
campaign (green dots in Fig. 5a), since the applicability of campaign averaged SPAR in $N_{CCN}$
calculations was confirmed by many former studies in the NCP (Deng et al., 2012; Wang et al., 2013;
Ma et al., 2016). During low RH periods, SPAR was generally lower than the campaign averaged
SPAR and the ratio between the calculated and measured $N_{CCN}$ were systematically higher (lasting for
the whole night). In summary, SA formation processes can induce significant deviation to $N_{CCN}$
prediction that varied with RH conditions and mainly resulted from the variation in MAF. Thus, for
accurate $N_{CCN}$ estimations, considering the variation of MAF (changes in the fraction of the
hygroscopic particles) is highly essential.

As SOA is generally considered to be more hygroscopic than POA (Frosch et al., 2011; Lambe

et al., 2011; Kuang et al., 2020a), the increase of hygroscopic particles or SA particles (both SIA and
SOA) were considered to be the cause for the increase of SPAR within 200 to 300 nm size range (Fig.
2). In order to account for the variations of hygroscopic particles or SA particles in $N_{CCN}$ calculation,
in the following part, Number Fraction of hygroscopic particles (GF(90%, 200 nm)>1.22, $NF_{hygro}$)
measured by HTDMA and Mass Fraction of SA particles ($MF_{SA}$) measured by ACSM in this campaign
were used to represent MAF variations and to provide calculation of $N_{CCN}$ at SS of 0.05% with smaller
deviations combined with PNSD measurement. To be noted, in order to highlight the application of
using $MF_{SA}$ as estimation of MAF variations on $N_{CCN}$ calculation, the campaign averaged $D_a$ from
SPAR curves was used.

Based on the bulk hygroscopicity derived from particle chemical compositions measurements

($\kappa_{chem}$), a critical diameter for CCN activation can be calculated based on κ-Köhler theory. With this
critical diameter, $N_{CCN}$(0.05%) can be predicted incorporating measured PNSD ($N_{CCN\_Chem}$).  κ of
accumulation mode particles derived from chemical composition of the bulk aerosol might bear
significant uncertainties, which leads to significant deviations of $N_{CCN}$ prediction. However, in practice,
chemical compositions measurements specifically for accumulation mode particles are not common,
thus bulk aerosol chemical compositions are commonly applied in CCN studies as substitute (Zhang
et al., 2014; Zhang et al., 2016; Che et al., 2017; Cai et al., 2018), especially when particle
hygroscopicity measurements were in lack. As can be seen in Fig. 6(a), $N_{CCN\_meas}$ at 0.05% SS was
strongly underestimated by $N_{CCN\_Chem}$, especially at lower $N_{CCN\_meas}$ ($\sim 10^2$ # cm$^{-3}$), which is similar to
the results of studies that encountered high fractions of organics (Chang et al., 2010; Kawana et al.,
2015). This deviation between $N_{CCN\_meas}$ and $N_{CCN\_Chem}$ may have resulted from the hypothesis of
internal mixing state and the difference of particle hygroscopicity derived by particle chemical
composition measurements and CCN activity. Fig. 6(b) depicts the correlation between mass fraction
of SA ($MF_{SA}$) and MAF at 0.05% SS. $MF_{SA}$ was generally positively correlated to MAF (r=0.8) with
slight underestimations, suggesting that externally mixed SA dominated CCN-active particles. Thus,
in the prediction of $N_{CCN}$, real-time SPAR can be calculated from campaign average $D_a$ and MAF
assumed to be equal to real-time $MF_{SA}$ ($N_{CCN\_MF}$). As displayed in Fig. 6(c), the underestimation and
correlation between $N_{CCN\_cal}$ and $N_{CCN\_meas}$ was improved after introducing $MF_{SA}$ into $N_{CCN}$ calculation.
Additionally, the diurnal variations of $N_{CCN\_cal}/N_{CCN\_meas}$ ratio based on different methods of $N_{CCN}$
calculation during the whole campaign were shown in Fig. 6(d). By considering real-time $MF_{SA}$
variations, the deviation of calculated $N_{CCN}$ (Real-time MF) can be reduced throughout the day,
compared to $N_{CCN\_Chem}$ (Real-time Chem). Meanwhile, using an averaged $MF_{SA}$ to estimate SPAR and
$N_{CCN}$ could also reduce deviations of calculated $N_{CCN}$ (Averaged MF), however, demonstrated a much
stronger diurnal variation than the deviation of $N_{CCN\_MF}$.

Based on the bulk hygroscopicity derived from GF measurements ($\kappa_{GF}$) at 200 nm, $D_a$ can be

calculated based on the κ-Köhler theory, which can be applied to predict $N_{CCN}$ at 0.05% SS ($N_{CCN\_GF}$)in
combination with measured PNSD. Fig. 7(a) reveals that $N_{CCN\_meas}$ were strongly underestimated by
$N_{CCN\_GF}$ (by more than 30%), which might have resulted from the hypothesis of internal mixing state
and the difference of particle hygroscopicity derived by GF and particle CCN activity measured under
different water vapor saturated conditions. Fig. 7(b) depicts the positive correlation between $NF_{hygro}$
and MAF at 0.05% SS, which was weaker than that between $MF_{SA}$ and MAF. Similar as before, $NF_{hygro}$
was applied as a proxy for MAF in the $N_{CCN}$ calculation, which also improved the underestimation and
correlation between $N_{CCN\_cal}$ and $N_{CCN\_meas}$ (Fig. 7(c)). Also, the campaign averaged $D_a$ in Fig. 5a.was
used to calculate SPAR curves and $N_{CCN}$. The diurnal variations of the $N_{CCN\_cal}/N_{CCN\_meas}$ ratio based
on different methods of $N_{CCN}$ calculation during the whole campaign are shown in Fig. 7(d). By
considering the real-time variation of $NF_{hygro}$, the deviation of $N_{CCN\_NF}$ (Real-time NF) was mainly
reduced during nighttime compared to $N_{CCN\_GF}$ (Real-time GF). Meanwhile, applying an averaged
$NF_{hygro}$ to estimate SPAR and $N_{CCN}$, reduced the deviations of calculated $N_{CCN}$ (Averaged NF) during
nighttime as well, but its deviations demonstrated stronger diurnal variations than those of $N_{CCN\_NF}$. If
GF-PDF were directly used to calculate $N_{CCN}$, $N_{CCN\_cal}$ would agree well with measured $N_{CCN}$ (Fig.
S4), because in this way the mixing state of aerosol would have been accounted for. However,
compared to the approach using GF-PDF, $NF_{hygro}$ is easier to apply in $N_{CCN}$ calculation and can yield
similar accuracies.

In summary, MAF exhibited strong diurnal variation that varied under different RH conditions

due to different SA formation mechanisms, which contributed most to $N_{CCN}$ estimation deviations if
unaccounted for. The diurnal variations of MAF at the five measured SSs (Fig. S5) revealed significant
diurnal variations at low SSs (0.05% and 0.07%) that were dependent on RH conditions, while only
small diurnal variations that were insensitive to the RH conditions were detected at SSs above 0.2%.
In general, MAF became lower at lower SSs, especially during nighttime. As the fraction of CCN-
active particles were generally hygroscopic and composed of secondary compounds, positive
correlation was found between MAF, $MF_{SA}$ and $NF_{hygro}$. Although a good prediction of $N_{CCN}$(0.05%)
was achieved by applying an averaged MAF (Figs. 5, 6d and 7d), in practice, this would still require
CCN measurements or HTDMA/chemical composition measurements as proxies. Additionally,
deviations of $N_{CCN\_cal}$ based on the averaged MAF can be large under low RH conditions (Fig. 5c),
while time-dependent MAF can eliminate a great part of these deviations. Thus, by replacing MAF
with real-time $MF_{SA}$ or $NF_{hygro}$ when deriving SPAR curves, the relative deviation of $N_{CCN}$(0.05%)
calculation can be reduced. The proposed $N_{CCN}$ parameterization using $MF_{SA}$ can also be easily
adopted by chemical-transport and climate models, improving their representation of $N_{CCN}$ changes
due to distinct SA formation processes.

**4. Conclusions**

SA formation drives the development of haze pollution in the NCP and can result in significant variations of PNSD and aerosol hygroscopicity. Studies in the NCP have shown that the mechanism of SA formation can be affected by relative humidity (RH), and thus has different influences on the aerosol hygroscopicity and PNSD under distinct RH conditions. The difference in particle size where SA formation is taking place and the different chemical compositions of formed SA can result in different variations of CCN activity. Thus, it is essential to study the influence of SA formation on CCN activity of existing accumulation mode particles under different RH conditions in the NCP. As $N_{CCN}$ is often predicted based on real-time PNSD and parameterized SPAR, the influence of varying SPAR in distinct SA formation processes on $N_{CCN}$ calculation needs to be evaluated in detail.

Based on the measurements of CCN-activity, particle hygroscopicity, particle chemical composition, PNSD during the McFAN campaign in Gucheng winter 2018, the influences of SA formation on CCN activity and $N_{CCN}$ calculation under different RH conditions were investigated especially at SSs lower than 0.07%. Two kinds of SA formation events were identified under different RH conditions with distinct variations in PM and $N_{CCN}$ at 0.05% SS. Under high RH conditions, which corresponds to the periods with minimum RH higher than 50% in daytime, strong SA formation and $N_{CCN}(0.05\%)$ enhancements with strong hygroscopic particles and SIA dominated contribution to SA (>70%) was found. While under low RH conditions, which corresponds to the periods with daytime minimum RH below 30%, moderate SA formation and $N_{CCN}(0.05\%)$ enhancements with moderately hygroscopic particles and SOA dominated contribution to SA was found. However, the increase of $N_{CCN}$ under the a same amount of SA formation was stronger under low RH conditions and weaker under high RH conditions. This was because the formation of SA under low RH conditions was more concentrated in particle size range smaller than 300 nm and added more mass to CCN-inactive particles turning them into CCN-active ones after SA formation.

In addition, strong diurnal variations of CCN activity of particles at 0.05% SS due to the strong SA formations were also observed, both varying with RH conditions. $N_{CCN}(0.05\%)$ was significantly underestimated when MAF (SPAR parameter) variations were not considered. As the fraction of CCN-active particles were generally hygroscopic and composed of secondary compounds, there were good correlation among MAF inferred from measurements of CCN activity, particle hygroscopicity and particle chemical compositions. Thus, the relative deviation of $N_{CCN}(0.05\%)$ estimation can be reduced by applying measurements of particle hygroscopicity or particle chemical compositions as a proxy for aerosol mixing state.

This study can further the understanding of the impact of SA formation on CCN activity and $N_{CCN}$ calculation, specifically for SA formations on existing particles, which can strongly affect cloud microphysics properties in stratus clouds and fogs. The investigation of the influence of SA formation on CCN activity of existing particles in this study is important for improving $N_{CCN}$ parameterizations in chemical-transport and climate models, so that they can account for the large variations induced by SA formation processes.

**Supporting Information**

The supporting information is available in a separate file.

**Data availability.**

The data used in this study are available from the file sharing link (https://pan.baidu.com/s/1iSMdEQj_KRrjmmrXtXNJUg) using extracting code db2p.

**Author contributions.**

JT, YK and NM designed this research. JT performed the data analysis and wrote the manuscript. YC, HS, NM, YK, JT, and JH planned this campaign. JT and YZ conducted the CCN measurements. YS and YH conducted the ACSM measurements and the ACSM PMF analysis. JH and QL conducted the HTDMA measurements. LX and YZ conducted the particle number size distribution measurements. WX conducted the measurements of CO and meteorological parameters. YC, HS, YS, YK and NM contributed to the revisions of this manuscript and all other coauthors have contributed to this paper in different ways.

**Acknowledgement**

We acknowledge the National Key Research and Development Program of China (grant no. 2017YFC0210104), the National Natural Science Foundation of China (grant no. 91644218 and 41805110), the Guangdong Innovative and Entrepreneurial Research Team Program (Research team on atmospheric environmental roles and effects of carbonaceous species: 2016ZT06N263), Special Fund Project for Science and Technology Innovation Strategy of Guangdong Province (2019B121205004) and the Basic Research Fund of CAMS (2020Z002)..

**Conflicts of interest**

There are no conflicts to declare.

629

630

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

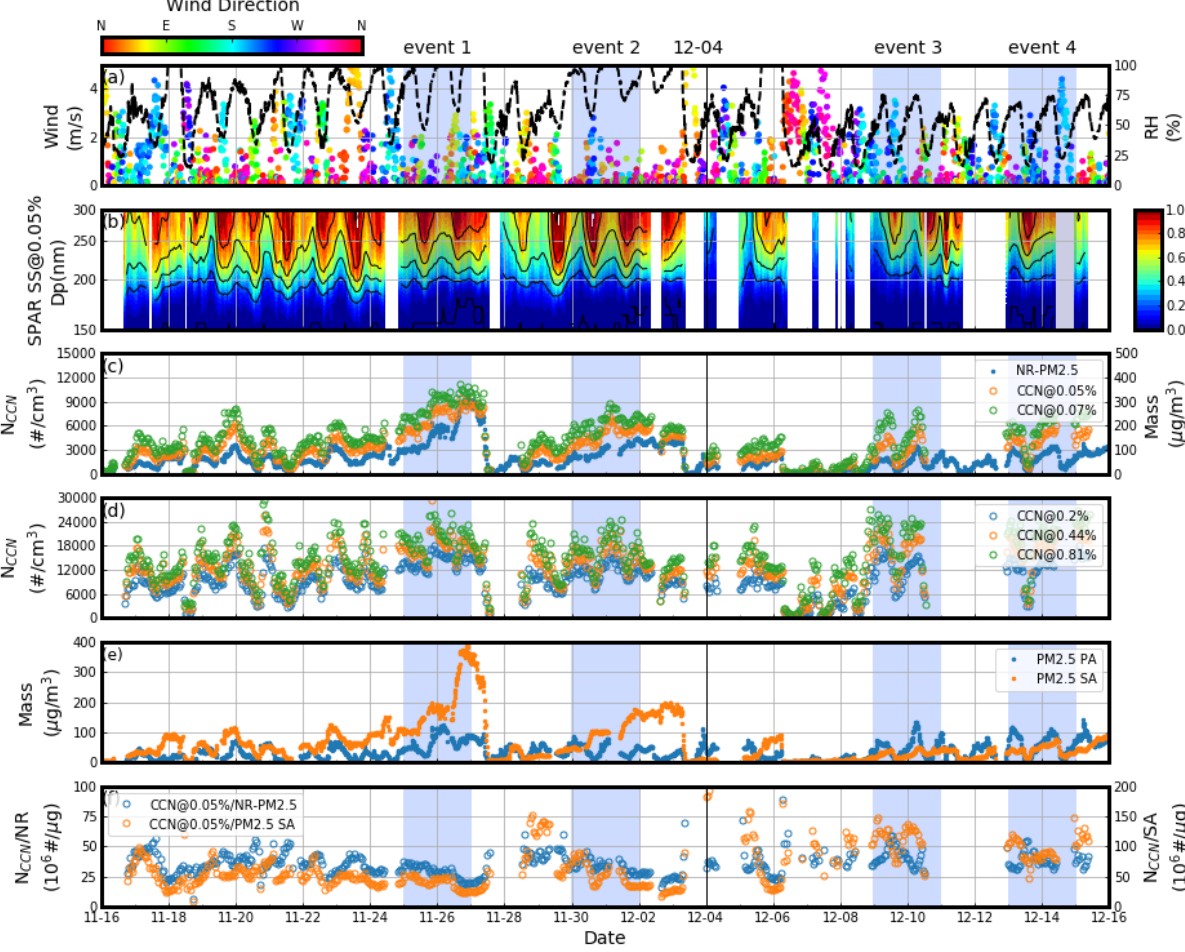


Fig 1. Overview of the measurements during the campaign: (a) dots represent wind speed with color
indicating wind direction, and black lines represent RH; (b) SPAR under SS of 0.05%; (c) blue, green
and yellow dots represent $N_{CCN}$ under SS of 0.05% and 0.07%, and mass concentration of NR-PM2.5,
respectively; (d) blue, green and yellow dots represent $N_{CCN}$ under SS of 0.2%, 0.44% and 0.81%,
respectively; (e) blue and yellow dots represent mass concentration of $PM_{2.5}$ PA and $PM_{2.5}$ SA
respectively; (f) blue and yellow dots represent ratio between $N_{CCN}$ and mass concentration of NR-
$PM_{2.5}$ and $PM_{2.5}$ SA, respectively. There were four events with significant enhancements of $N_{CCN}$
during the blue shaded periods.

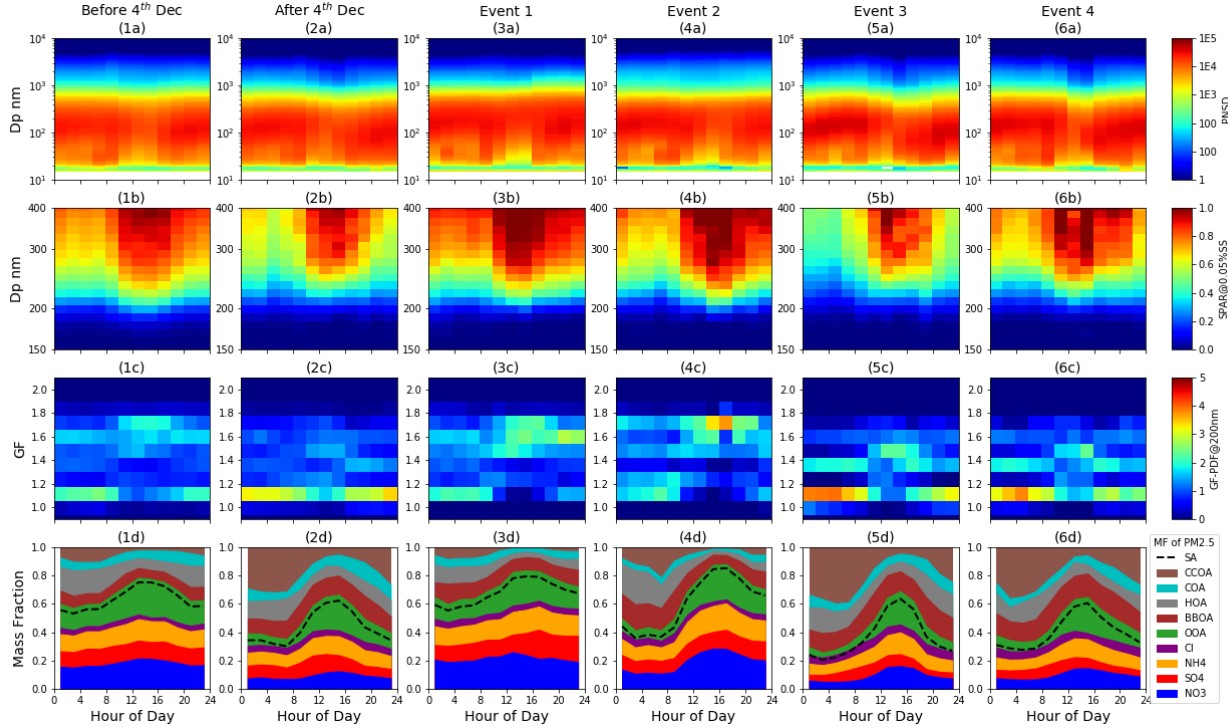


Fig 2. Diurnal variation of (a) PNSD, (b) SPAR at SS of 0.05%, (c) GF-PDF at 200 nm and (d) mass
fraction of different PM$_{2.5}$ chemical species during high RH periods before 4$^{th}$ Dec (1), low RH
periods after 4$^{th}$ Dec (2) and the four events (3-6), including OA factors: hydrocarbon-like OA
(HOA), cooking OA (COA), biomass burning OA (BBOA), coal combustion OA (CCOA), and
oxygenated OA (OOA).

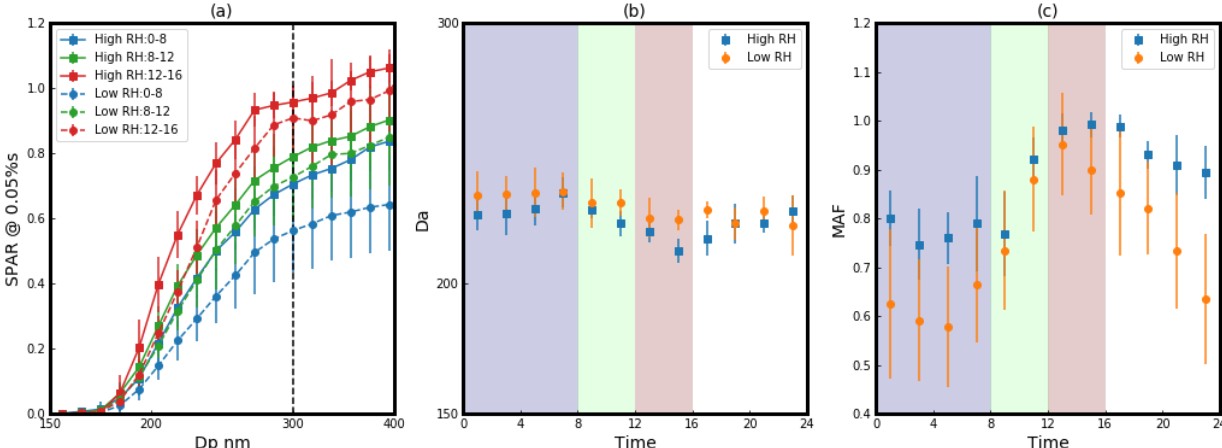


Fig 3. (a) The averages of SPAR curves at SS of 0.05% in three different time periods (blue: 0:00-
8:00; green: 8:00-12:00; red: 12:00-16:00) during high (squares with solid line, event 1 and 2) and low
(dots with dashed line, event 3 and 4) RH events. Diurnal variation of (b) $D_a$ and (c) MAF under high
(blue) and low (yellow) RH conditions. The blue, green and red shades correspond to with the three
periods in (a & d). Error bars indicate the standard deviations of data.

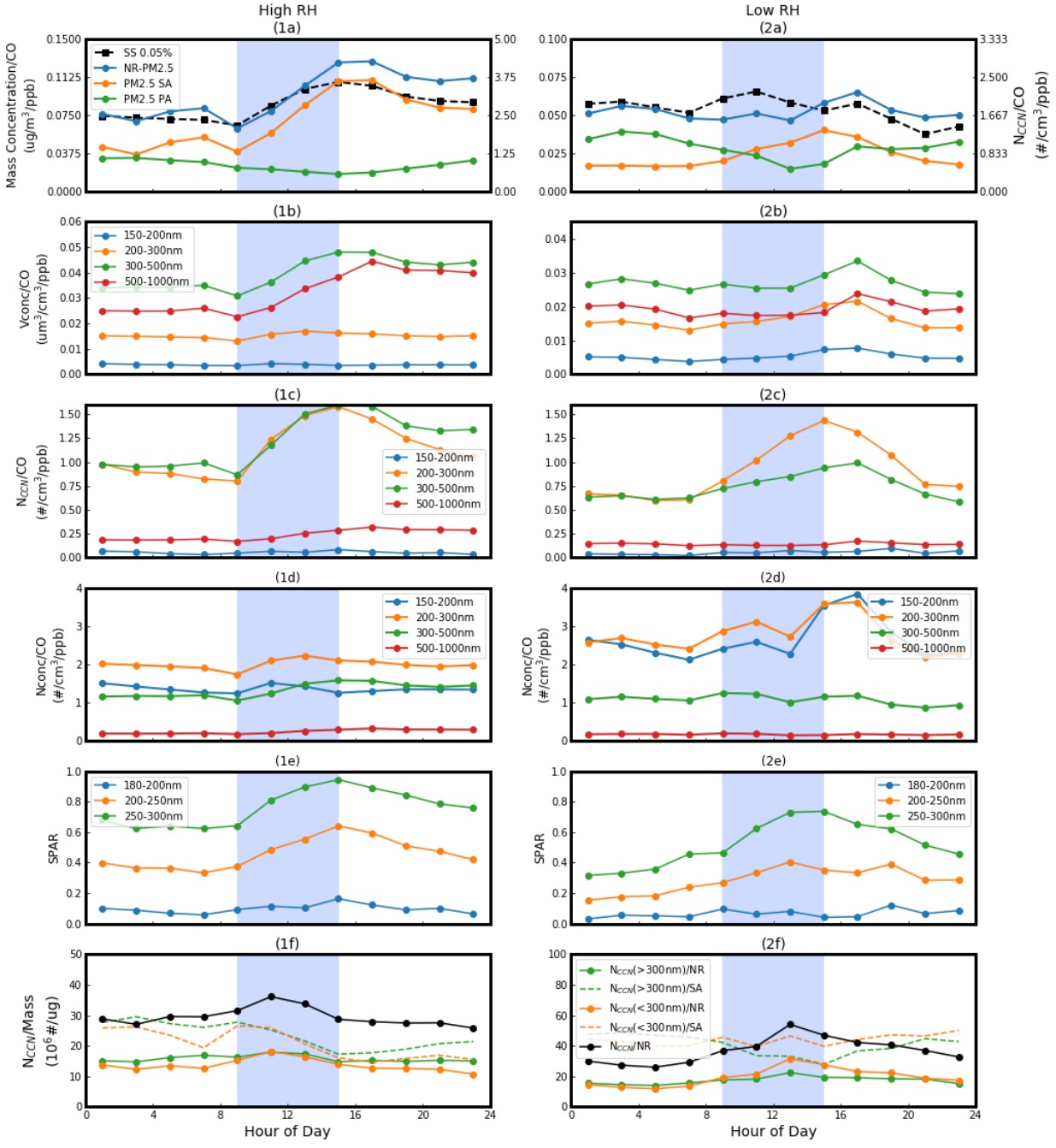


Fig 4. During different RH events, the average diurnal variation of (a) the ratios between particle
mass concentration (dots with solid lines; blue: NR-PM$_{2.5}$; yellow: PM$_{2.5}$ SA; green: PM$_{2.5}$ PA) and
CO concentration, and the ratio between N$_{CCN}$ at SS of 0.05% and CO concentration (squares with
solid line); (b) the ratios between particle volume concentration (Vconc) of different particle size
range (indicated by colors) and CO concentration; (c) the ratios between N$_{CCN}$ of different particle
size range at SS of 0.05% (indicated by colors) and CO concentration; (d) the ratios between particle
number concentration (Nconc) of different particle size range (indicated by colors) and CO
concentration; (e) SPAR of different particle size range (indicated by colors); (f) the ratios between
N$_{CCN}$ at SS of 0.05% (black: bulk N$_{CCN}$; yellow: N$_{CCN}$ with particle size larger than 300 nm; blue:
$N_{CCN}$ with particle size smaller than 300 nm) and mass concentration of NR-PM$_{2.5}$ SA and the ratios
between $N_{CCN}$ and mass concentration of NR-PM$_{2.5}$ (dashed lines).

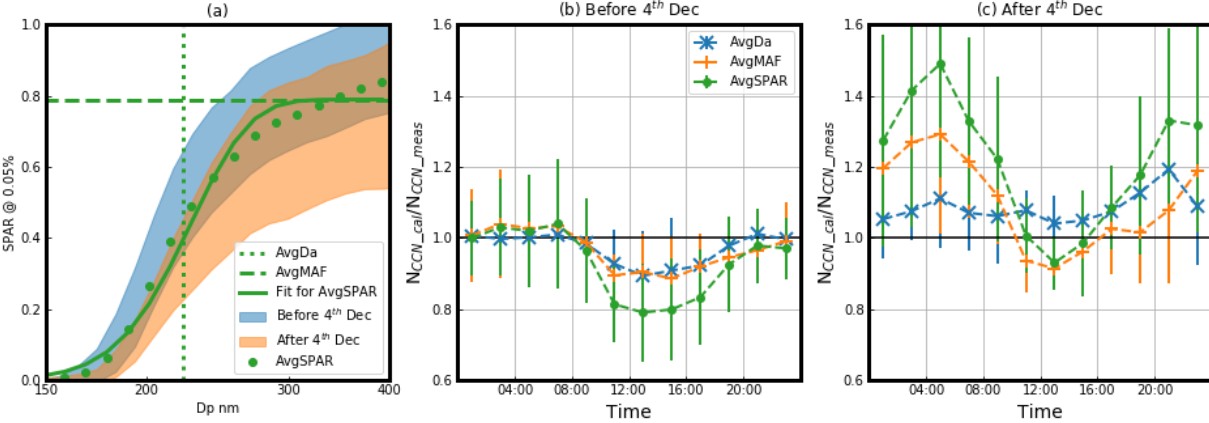


Fig 5. (a) The averaged SPAR at SS of 0.05% during the campaign (green scatters), the
corresponding fitting curve (green line) and the averaged fitting parameters (dotted line for $D_a$ and
dashed line for MAF). The blue and yellow shaded areas represent the variations of SPAR before 4th
Dec and after 4th Dec, respectively. The ratio between calculated $N_{CCN}$ and measured $N_{CCN}$ under (b)
before and (c) after 4th Dec. Bars represent one standard deviation and colors represent different
calculation of SPAR curves: green represent average SPAR during the campaign (AvgSPAR),
yellow represent SPAR calculated with average $D_a$ and real-time MAF (AvgDa) and blue represent
SPAR calculated with average MAF and real-time $D_a$ (AvgMAF).

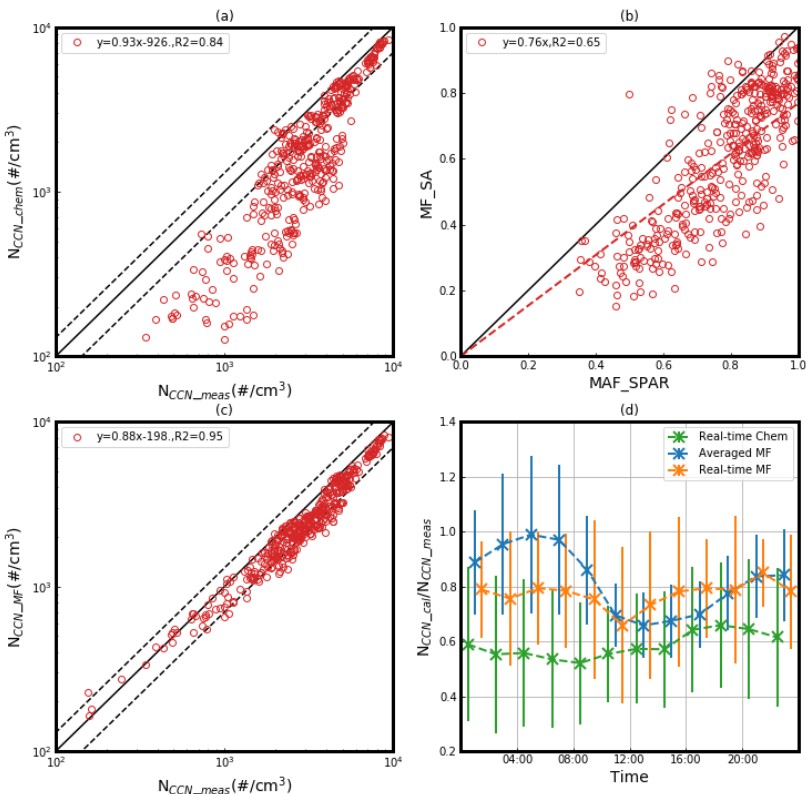

960

Fig 6. (a) The comparison between calculated $N_{CCN}$ based on $\kappa$ derived from bulk particle chemical compositions ($N_{CCN\_chem}$) and measured $N_{CCN}$ at SS of 0.05%. (b) The correlation between MAF and mass fraction of secondary aerosol ($MF_{SA}$). (c) the comparison between calculated $N_{CCN}$ based on SPAR derived from real-time $MF_{SA}$ and average $D_a$ ($N_{CCN\_MF}$) and measured $N_{CCN}$. The black dashed lines represent the relative deviation of 30%. (d) the diurnal variations of the ratio between the calculated and measured $N_{CCN}$ during the whole campaign based on different methods (green: $N_{CCN\_chem}$; blue: $N_{CCN}$ calculated based on SPAR derived from averaged $MF_{SA}$ and average $D_a$; yellow: $N_{CCN\_MF}$).

969

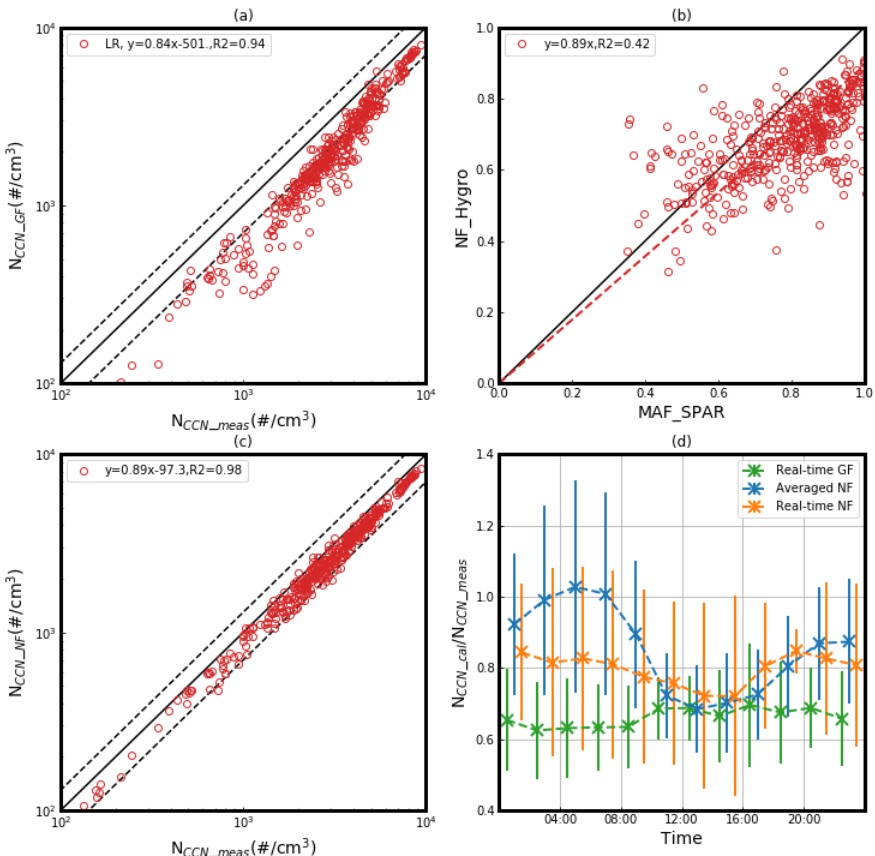

970

Fig 7. (a) The comparison between calculated $N_{CCN}$ based on κ derived from bulk GF at 200 nm ($N_{CCN\_GF}$) and measured $N_{CCN}$ at SS of 0.05%. (b) The correlation between MAF and number fraction of hygroscopic particles ($NF_{hygro}$, GF>1.2). (c) The comparison between calculated $N_{CCN}$ based on SPAR derived from real-time $NF_{hygro}$ and average $D_a$ ($N_{CCN\_NF}$) and measured $N_{CCN}$. The black dashed lines represent the relative deviation of 30%. (d) the diurnal variations of the ratio between the calculated and measured $N_{CCN}$ during the whole campaign based on different methods (green: $N_{CCN\_GF}$; blue: $N_{CCN}$ based on SPAR derived from averaged $NF_{hygro}$ and average $D_a$; yellow: $N_{CCN\_NF}$).