# Peer review of "Secondary aerosol formation alters CCN activity in the North China"

_Atmospheric Chemistry and Physics, 2020_

## Referee Comment (RC1) · Anonymous Referee #2 · 12 Nov 2020

In the manuscript "Secondary aerosol formation alters CCN activity in the North China Plain", the authors conducted a field study in North China Plain and investigated the influence of second aerosol (SA) formation on CCN activity and on the calculated CCN number concentrations derived from particle number size distribution (PNSD). The CCN activity at 0.05% supersaturation (SS) was discussed. The authors focused on CCN activation at low SS where mainly accumulation mode particles act as CCN and thus on cases of SA in the presence of accumulation mode particles. They found that at two different RH, SA formation had different influence on CCN activity of aerosols. At high RH (minimum RH>50%), SA mass mostly added to larger particles (>300 nm), which resulted in weaker enhancement of CCN activity for per SA mass added as these larger particles were already CCN-active before SA formation. At low RH (minimum RH<30%), SA mass mostly added to smaller particles (<300 nm), which resulted in stronger enhancement of CCN activity for per SA mass added as smaller particles were not CCN-active before SA formation. In addition, they parameterized maximum activation fraction (MAF) using the correlation of MAF with hygroscopic particle number fraction or with mass fraction of SA. The calculated CCN concentrations derived from PNSD using parameterized MAF, campaign average activation diameter and width of activation curve matched better with measured ones compared with using PNSD and kappa from either chemical composition or hygroscopic growth.

How aerosol formation and growth affect CCN activity is an important question. The manuscript provides valuable case study on how secondary aerosol formation influence CCN activity for low stratus clouds and fogs. This study carried out comprehensive measurement of aerosol related to CCN activity/hygroscopicity. The findings are interesting. However, I have some comments about the manuscript to address before it is considered for publishing in ACP.

Major comments:

1. The manuscript only discussed the results at 0.05% SS. How do the findings depend on SS? What about the results at other SS such as 0.1% and 0.2% SS, which is also typical for low stratus clouds? In addition, I suggest explicitly specifying SS when CCN activity or CCN concentration is discussed.

2. I was somewhat surprised to notice that MAF only reached 0.4-0.6 in Fig. 3. Why were the data larger than 300 nm excluded (L148)? Did the activation fraction reach around one at larger sizes? If one fit Eq.7 to e.g. blue curves in Fig. 3a only till 300 nm, the $D_a$ derived at half MAF might be incorrect.

3. The authors reported two cases at high RH and only one case at low RH. It would be helpful to discuss how general these conclusions are regarding the influence of SA on CCN activity. The authors seem to indicate that RH is the dominant factor. What about other conditions? For example, how would the size and chemical composition of existing particle affect the conclusion here?

4. I had some difficult time reading the manuscript. I suggest the authors streamlining the writing substantially. Additionally, there are numerous language problems. For example, in many cases, a space was missing before a unit. More specific problems are listed below.

Specific comments

1. L254, it is half MAF that can represent the number fraction of CCNs to total particles at particle size around Da. Also "represents" should be "represent".

2. L264, how are PA and SA characterized?

3. L276, how are the time ranges of these events defined? By $PM_{2.5}$ concentration?

4. L283-286, for me it is hard to tell from the data that the ratios were really lower after 4[th] Dec. Nor can I discern the "decreasing trends".

5. L294, by "the increase of hygroscopic particles", do you mean number or mass concentration?

6. L304, is this statement necessarily true?

7. L311, by which metric do you define "CCN activity"? Do you refer to activation fraction?

8. L313, "the enhancement of particle CCN activity was stronger in low RH events", which metric or data is this statement based on?

9. L319, it is not obvious to tell if there is "the increases of Da".

10. L321, again "the enhancement of CCN activity was lighter", what metric or data is this statement based on?

11. L325-327, "unchanged CCN activity at low RH conditions", how is this statement drawn? Is this finding also valid for other SS?

12. L339, "relatively smaller variations of particle density", this needs support from data or literature.

13. L365, "decreased continuously", it seems not to be a continuous decrease.

14. L445-447, it is not surprising that the correlation of $N_{CCN\_chem}$ with $N_{CCN\_meas}$ was not good as kappa was only derived from chemical composition of the bulk aerosol, which is highly biased to larger particles.

15. L439-440, such a statement is not necessarily true. Primary particles can be CCN active. In addition, the authors defined kappa>0.1 as hygroscopic particle in the method part. Kappa of SOA can be <0.1, which contracts the statement here.

16. L454, "real-time MAF can be estimated by MF_SA", how to estimate, by simple linear regression?

17. L473, how do MAF and its diurnal variation depend on SS?

18. L495, in the abstract, 50% was used while here 40% was used...

19. L509, "mixing state", is the right word here? What is the mixing state of these aerosols based on the "measurements of CCN activity, particle hygroscopicity and particle chemical compositions"?

20. L797, in Fig. 2, it is helpful to describe the OA factors in the method part.

Technical comments:

1. L214, "$NF_{hygro}$" was written as "NF_hygro" later.

2. L272, "Dec." should be "Dec".

3. L283, "are" should be "is". "Higher" might be better than "stronger".

4. L324, "um" should be "μm".

5. L346, "normalized" is missed before $PM_{2.5}$? "Fig. 4(a1)" should be "Fig. 4(1a)".

6. L356, "of" should be omitted.

7. L376-378, this sentence is hard to understand.

8. L432-433, "there were similar difference between CCN_AvgMAF and $N_{CCN\_meas}$" this sentence is hard to understand.

9. L452, "the application of MF_SA on NCCN calculation were shown", it is in Fig. 6c rather than 6b.

10. L467-468, this sentence is hard to understand.
11. L825, "the" should be "The".

---

## Referee Comment (RC2) · Anonymous Referee #1 · 5 Jan 2021

The manuscript is about the influence of the secondary aerosol (SA) formation on the CCN activity based on a measurement campaign done on the North China Plane. The topic is very interesting, I would very much like to see a thorough study on it to get published. However, as the manuscript is now prepared, I have doubts about its quality, and in this form I cannot recommend it to be published in ACP. It needs a serious and thorough rework based on the referees' comments before it can be cosidered to be published. Please find my comments and remarks in the following.

General comments:

1. Too few events were analyzed in my opinion, to see whether really the change in RH cause a different CCN behavior. For such a study, more data would be needed than two short events for the high RH period and a single event for the low RH period. At least

use as many days for the data analysis as possible from this data set. For me, it looks like that you have chosen your RH criteria such, that only those days are included that you want to analyze even if there would be the possibility to include many more days when the RH was high or low. E.g. why don't you use 14th of December as a low RH event? Either use almost all the days with higher RH and lower RH for this comparison or do not do this low and high RH separation at all. Compare the campaign averages before and after the 4th of December, something like you show in Figure S2. As it is presented now, I am not convinced, that there is a significant difference between the low and high RH cases based on a solely 3 events. What if during the single low RH event something else than the RH caused the difference in the CCN activity? How can you be sure, that the RH is responsible?

2. Why do you only show the results at SS=0.05% when you have measured at 5 different SSs? Please show all the supersaturations you have measured. You could generally try to speculate a little bit less in the paper and at the same time show more important data, if you are afraid, that the paper will be too long. I know that you have mentioned, that you would like to focus on the low SS case, but you have still two other measured SSs smaller or equal then your upper limit of SS of interest (0.2%). Please at least include them in this paper. It would be nice to see whether SA formation have an effect on the CCN activity at those higher SSs as well or not.

3. At many parts of the paper, the MAF (maximum activated fraction) parameter appears (together with a single sigmoid fit) and is used for the fraction of the hygroscopic particles. As I mention later in the detailed comments, this parameterization/fit can only be used in certain cases. You should include a discussion and provide information on how well this fit could be used for your data. And dependent on the SS set in the CCNC, the MAF you present has a different meaning. You only show measurements at SS=0.05%, at this SS and with the highest considered dry diameter of 300nm, this MAF has the meaning of the fraction of the particles having a kappa at least approx. 0.22, far far away from non-hygroscopic. 1/3 AS and 2/3 BC would have such a kappa.

Use MAF accordingly, and correctly! And I would need proof that this fitting method can be used for your data at any time during the campaign. For the 0.05% case, it assumes that there are no particle present around the kappa of 0.22, just a population with much higher kappa and a population with significantly lower kappa. Was it the case for the whole measurement period? If there will be other SSs included in the paper following my suggestion, then please check and show what the MAF would mean at that SS, like e.g. at 0.2% and maximum diameter of 300nm, the MAF would be the fraction of the particles having a kappa higher than approx. 0.013. Or a much better choice would be doing such a fit until a constant kappa at different SSs which would mean different maximal fit diameters. That would have a more useful meaning. Like the fraction of the particles having a hygroscopicity below kappa 0.1. That would mean that you have to use the measurements until a higher diameter than 300nm (approx. 390nm) at SS=0.05% which you did not include because of having too much noise. But that problem could be solved following another one of my previous suggestions and using more data and doing some time averaging. You have many choices, choose something which you like, but it is very confusing right now, and this MAF, as calculated now, is not representative for the fraction of the non-hygroscopic particles.

4. Something is strange for me for Figure 5a. How can it be, that the ratio between the calculated and measured N_CCN is systematically below 1? I would expect using the partly or completely averaged SPAR (whichever trace I look at it), that the ratio is scattered around one, but not being always below (like in Figure 5b). For me, this could only happen if e.g. you have a systematic error in the fitting procedure, which always underestimate the measured SPAR, or something else. In my opinion something can not be correct here. Please explain me, if the data is correct, how that can be.

Detailed comments:

Line72: "different with those" did you mean here different from those?

Line 86: hydrophobic is a too strong expression here, I guess you mean nonhygroscopic

Line 132: how far was the container form the building of the gas measurements?

Line 139: you mention here the SS and the corrected SS of the CCNC, what is this correction? If it is simply the SS calibration, then you do not need to mention the wrong SS levels, just state the correct ones you determined based on the instrument calibration.

Line 154-156: about the inversion and multiple charge correction of the scanning CCNC system: you mention that a multiple charge correction was done and show some references, where details about it can be found. However, I really had to search longer among those papers until I found a method in one of them. Since the main result what you show in this paper is the SPAR, to my opinion the method of inversion/multiple charge correction has to appear a bit more detailed in this paper. And as I understood from the method I found in one of the references (if I found the method you used here), a simple correction only taking the multiple charged particles into account was applied. The width of the DMA transfer function was neglected. Please at least speculate on it, how much error you introduce to your measurement with this assumption.

Line 170 and 173: "under RH of 90%" please change under to at, under could be also understood as below

Line 179: you mention 4 dry sizes in Line 178 and then 6 sizes in this line. Which one is correct?

Line 209: What function was used for the fit?

Line 211: "(HGF?)" Typo?

Line 217: "(Da_hygro)" what is that?

Line 224: "dominate" change to dominant

Line 240: please change "reported in the same..." to "reported from the same..."

Line 243: what are the kappa values you used for the inorganics? The kappa theory is not a perfect parameterization of the water activity, and therefore it is not granted that a kappa you calculate from a HTDMA will be the same as what you get from a CCN measurement. For example, AS has a different kappa at supersaturation and at 90% RH. How did you take this into account? Please comment on it. And at what RH was the relationship for kappa_org determined in the mentioned study? Line 248: it is not generally parameterized, often but for sure not generally, please correct Line 249: change hydrophobic to non-hygroscopic, or what kind of hydrophobic particles do you mean? I am not aware of any kind of atmospheric aerosols that are hydrophobic. To my knowledge non-hygroscopic (kappa=0) aerosol particles activate like a completely non-soluble but wettable surface according to the Kelvin-effect. Hydrophobic particles activate at even worse than those, so at a higher SS.

Next to that, a CCNC can theoretically measure non-hygroscopic activation at any SS, you simply need to get to a high enough particle diameter. So please change the sentence accordingly mentioning, that your used setup, which only goes up to 300nm, was not able to capture the activation. Next to it, at your highest SS of 0.8%, non-hygroscopic particles (kappa=0) that have larger dry diameter than 270nm already activate. So, at your highest SS and diameter of 300 nm you should activate the non-hygroscopic particles as well and get an MAF of 1 (assuming now a very narrow DMA transfer function which might not be the case) independent on the fraction of the non-hygroscopic particles. Line 254: "can represents" do you mean here can represent or represents? The later would not be true, if you have a hygroscopic fraction of the aerosols with not a single kappa but a broader kappa distribution. Please include a discussion on this here.

Line 255: sigma of the error function: does not only include the heterogeneity of the hygroscopicity but also the transfer function of your measurement system, mainly the DMA transfer function.

Line 257: see my previous comments on "hydrophobic". Kappa<0.1 is not even nonhygroscopic. It would be something like a particle consisting of approx. 17% of AS and 83% of BC. One definitely cannot call this hydrophobic.

Line 266: change "is" to "was"

Line 290: you show the CCN activation ratio/fraction not activity Section 3.2: From figure 1 it looks like, that you have a strong diurnal variation of the CCN activity almost every day. Somehow you only show the results of the few selected events. Please show at least an average (and the variation) of all the days for the data you show in Figure 2. And discuss them. It would be also nice to show the diurnal variation of the number size distribution as well.

Line 301: please correct "hydrophobic"

Figure 3a: please include the standard deviation of the averages for the SPAR curves as error bars or shading

Figure 3b-c: what are the error bars? The error of the fits, or the standard deviations of the calculated averages, or something else?

Line 327-329: I do not understand this sentence

Line 331-33: I do not understand either

Line 348: "to can be expected" typo

Figure 4b: showing the number of aerosol particles instead of the volume would be much useful, the CCN activity is also measured by the number and not by the volume

Line 399-402: Sentence too long, please start a new sentence after "respectively" and reformulate if, it is hardly understandable.

Line 411: change please "was" to "is"

Line 420: do you mean "is calculated based on. . ."?

Line 420: CCN activity is not a quantity, somehow you use that through the whole paper

as it was. Please correct it everywhere. What do you mean by it here? The SPAR? Or some kind of N_CCN? How is the N_CCN_cal exatly defined? Or is that the calculated N_CCN? Please rewrite this whole sentence and explain how you exactly calculated the CCN prediction.

Line 426: "as to" -> "to as"

Line 439-442: For me it would be strange if using a completely different instrument for a kappa measurement from bulk chemistry assuming internally mixed aerosols would improve the N_CCN prediction compared to the prediction based on the averaged SPAR. Please do not introduce this prediction method as an improvement.

Line 440: please include the exact definition of the number fraction of hygroscopic particles!

Line 453: calling Rˆ2=0.59 a "strong correlation" is maybe a little bit too strong.

Figure 6: Please show the calculated vs. measured N_CCN for the methods you used for Figure 5 as well to have a comparison.

Line 459-472: you could not only use the bulk HTDMA hygroscopicity but the complete GF-PDF for the N_CCN estimation considering the mixing state of the aerosols as well. For sure, that would improve the calculation as well.

Line 473-478: If you want to show the importance of the changing MAF in the N_CCN prediction then you do not need all these calculations using the HTDMA and the AMS and the MAF prediction based on a whatever measured parameter of these instruments. Just simply show the calculated N_CCN (averaged MAF) vs the measured N_CCN (MAF as it was measued) as you calculated for the orange line in Figure 5. And as it looks like from Figure 5 you would not have an average error higher than 10% using the averaged MAF, so I am really not convinced about your summary statement. It might be important to take an average MAF different from 1 into account, but most probably not its time variation.

---

## Author Comment (AC1) · 22 Mar 2021

The comment was uploaded in the form of a supplement:
https://acp.copernicus.org/preprints/acp-2020-939/acp-2020-939-AC1-supplement.pdf

---

## Author Comment (AC2) · 22 Mar 2021

The comment was uploaded in the form of a supplement:
https://acp.copernicus.org/preprints/acp-2020-939/acp-2020-939-AC2-supplement.pdf

---

## Author Response (AR1)

Dear Editor,

We greatly thank the reviewers for their detailed review. Responses addressing reviewers' comments point-by-point were uploaded (and also attached to this file). The manuscript has been revised and improved accordingly.

Best Regards

Jiangchuan Tao and Nan Ma

Reviewer #1:

*The manuscript is about the influence of the secondary aerosol (SA) formation on the CCN activity based on a measurement campaign done on the North China Plane. The topic is very interesting, I would very much like to see a thorough study on it to get published. However, as the manuscript is now prepared, I have doubts about its quality, and in this form I cannot recommend it to be published in ACP. It needs a serious and thorough rework based on the referees' comments before it can be considered to be published. Please find my comments and remarks in the following.*

Response: Thanks for your comments. Suggestions and comments are addressed point-by-point and corresponding responses are listed below.

*General comments:*

*1. Too few events were analyzed in my opinion, to see whether really the change in RH cause a different CCN behavior. For such a study, more data would be needed than two short events for the high RH period and a single event for the low RH period. At least use as many days for the data analysis as possible from this data set. For me, it looks like that you have chosen your RH criteria such, that only those days are included that you want to analyze even if there would be the possibility to include many more days when the RH was high or low. E.g. why don't you use 14th of December as a low RH event? Either use almost all the days with higher RH and lower RH for this comparison or do not do this low and high RH separation at all. Compare the campaign averages before and after the 4th of December, something like you show in Fig. S2. As it is presented now, I am not convinced, that there is a significant difference between the low and high RH cases based on a solely 3 events. What if during the single low RH event something else than the RH caused the difference in the CCN activity? How can you be sure, that the RH is responsible?*

Response: Thanks for your comments.

Regarding this study, the statement that RH caused variations of CCN behavior is inaccurate, which may be due to some misleading statements in the original manuscript. In this study, our main point is that different SA formations during high RH and low RH environments are responsible for the variations of CCN activity. The "high (or low) RH events" is used to refer to the SA formation events under high (or low) RH conditions for convivence. As reported by Kuang et al., (2020), SA formation mechanisms and the corresponding influence on PNSD and particle chemical compositions are different during periods with different RH conditions. Thus, we investigated the variations of CCN activity measured during the same campaign and found that different SA formations can largely influence CCN activity due to variations of PNSD and particle chemical compositions. The misleading statements in the manuscript have been revised accordingly:

1. After the first sentence in Sec. 3.2. (discussing the Fig. 2), a description has been added as "*To be noted, the "high (or low) RH events" is used to refer to the SA formation events under high (or low)*

*RH conditions for convivence, and it doesn't mean that RH caused variations of CCN behavior."*

2. The first sentence of the second paragraph in Sec. 3.2 (discussing the Fig. 3a) has been revised as "*In Figs. 3a, detailed comparison of particle CCN activity during SA formation events of NCCN enhancements under different RH conditions are shown as the variations of SPAR curves.*"

3. The second sentence of the second paragraph in Sec. 3.3 (discussing the Fig. 5) has been revised as "*In former discussions, CCN activity (indicated by SPAR) revealed significant diurnal variations during this campaign, which were different during SA formations under distinct RH conditions.*"

4. The first sentence of the last paragraph in Sec. 3.3 (the summary of this section) has been revised as "*In summary, MAF exhibited strong diurnal variation that varied under different RH conditions due to different SA formation mechanisms, which ...*"

Following the reviewer's suggestion, the entire measurement period is split into a higher RH and lower RH parts, and the CCN activity and other measured parameters are compared (Figs. 2 (1a-1d) and (2a-2d)). Another low-RH episode ($13^{rd}$ Dec-$15^{th}$ Dec) has been also added (Fig. 2(6a-6d)). As can be seen in the revised Fig. 2, different variations of SPAR to SA formations can be found during the periods with different RH conditions. The variations of SPAR, GF-PDF and mass fraction of particle chemical compositions during the periods of high (or low) RH conditions were similar but less significant, as those during high-RH events 1 and 2 (or low-RH events 3 and 4). The four specific events (adding the $14^{th}$ Dec as an events under low RH conditions) with significant variations of CCN activity during SA formation are analyzed as examples. These events were chosen based on not only the RH but also the enhancement of SA. During event 3, the wind speed was generally low, the RH followed a general diurnal variations and SA mass grew steadily and continuously. Thus the interference of the variations of air mass and short-term local emissions can be eliminated and the influence of SA formation can be highlighted. While for event 4 ($14^{th}$ Dec), the increase of SA mass concentration was not so significant during the daytime and the windspeed was higher, suggesting stronger influence of other factors and less significant influence of SA formation. We have added corresponding discussion into the first paragraph of section 3.2 as follow:

"*The diurnal averages of PNSD, SPAR at SS of 0.05%, GF-PDF for 200 nm particle and mass fraction of particle chemical compositions during high RH periods before $4^{th}$ Dec, low RH periods after $4^{th}$ Dec and the four events are shown in Fig. 2, respectively. To be noted, ... CCN behavior. As can be seen in Figs. 2 (1b) and (2b), different variations of SPAR due to SA formations can be found during the periods with different RH conditions. The average diurnal variations of these parameters for the entire high RH stage and low RH stage as shown in Figs. 2 (1a-1d) and (2a-2d) revealed similar but more smoothed variations as in the four selected events. The four events are discussed and intercompared in the following to magnify the differences under distinct RH conditions.*"

We have also added corresponding choosing criteria for the events into the section 3.1 as follow:

"*These events were selected based on the similarity of $PM_{2.5}$ concentration and evolution, while*

*the time window was fixed to two days for the convenience of intercomparing. In addition, during these events, the wind speed was generally low, the RH followed a general diurnal variations and SA mass grew steadily and continuously. Thus the interference of the variations of air mass and short-term local emissions can be eliminated and the influence of SA formation can be highlighted.*"

In addition, we have also revised Fig. 1 (shown later in comment 2), 3 and 4 (shown below) accordingly, and the corresponding results in these figures are still valid. And as mentioned later in comment 2, variations of CCN activity at SSs of 0.07% and 0.2% during SA formation events including event 4 are also shown in Fig. S2 in the supplements.

[Figure]

**Fig 2. Diurnal variation of (a) PNSD, (b) SPAR at SS of 0.05%, (c) GF-PDF at 200nm and (d) mass fraction of different PM₂.₅ chemical species during high RH periods before 4ᵗʰ Dec (1), low RH periods after 4ᵗʰ Dec (2) and the four events (3-6), including OA factors: hydrocarbon-like OA (HOA), cooking OA (COA), biomass burning OA (BBOA), coal combustion OA (CCOA), and oxygenated OA (OOA).**

[Figure]

**Fig 3. (a) The averages of SPAR curves at SS of 0.05% in three different time periods (blue: 0:00-8:00; green: 8:00-12:00; red: 12:00-16:00) during high (squares with solid line, event 1 and 2) and low (dots with dashed line, event 3 and 4) RH events. Diurnal variation of (b) D$_a$ and (c) MAF under high (blue) and low (yellow) RH conditions. The blue, green and red shades correspond to with the three periods in (a & d). Error bars indicate the standard deviations of data.**

[Figure]

**Fig 4. During different RH events (1: event 1 and 2; 2: event 3 and 4), the average diurnal variation of (a) the ratios between particle mass concentration (dots with solid lines; blue: NR-PM2.5; yellow: PM2.5 SA; green: PM2.5 PA) and CO concentration, and the ratio between NCCN at SS of 0.05% and CO concentration (squares with solid line); (b) the ratios between particle volume concentration (Vconc) of different particle size range (indicated by colors) and CO concentration; (c) the ratios between NCCN of different particle size range at SS of 0.05% (indicated by colors) and CO concentration; (d) the ratios between particle number concentration (Nconc) of different particle**

**size range (indicated by colors) and CO concentration; (e) SPAR of different particle size range (indicated by colors); (f) the ratios between NCCN at SS of 0.05% (black: bulk NCCN; yellow: NCCN with particle size larger than 300nm; blue: NCCN with particle size smaller than 300nm) and mass concentration of NR-PM2.5 SA and the ratios between NCCN and mass concentration of NR-PM2.5 (dashed lines).**

*2. Why do you only show the results at SS=0.05% when you have measured at 5 different SSs? Please show all the supersaturations you have measured. You could generally try to speculate a little bit less in the paper and at the same time show more important data, if you are afraid, that the paper will be too long. I know that you have mentioned, that you would like to focus on the low SS case, but you have still two other measured SSs smaller or equal then your upper limit of SS of interest (0.2%). Please at least include them in this paper. It would be nice to see whether SA formation have an effect on the CCN activity at those higher SSs as well or not.*

Response: Thanks for your suggestions. We have added the variations of CCN number concentration at the five measured SSs into Fig. 1, the variations of SPAR and the ratios between CCN number concentration and PM2.5 at SS of 0.07% and 0.2% in Fig. S1 and the diurnal variations of SPAR at SS of 0.07% and 0.2% in Fig. S2, as follow:

[Figure]

**Fig 1. Overview of the measurements during the campaign: (a) dots represent wind speed with color indicating wind**

**direction, and black lines represent RH; (b) SPAR under SS of 0.05%; (c) blue, green and yellow dots represent NCCN under SS of 0.05% and 0.07%, and mass concentration of NR-PM2.5, respectively; (d) blue, green and yellow dots represent NCCN under SS of 0.2%, 0.44% and 0.81%, respectively; (e) blue and yellow dots represent mass concentration of PM2.5 PA and PM2.5 SA respectively; (f) blue and yellow dots represent ratio between NCCN and mass concentration of NR-PM2.5 and PM2.5 SA, respectively. There were four events with significant enhancements of NCCN during the blue shaded periods.**

As the Fig. 1 shows, the variations of NCCN at 0.07% were similar to those at 0.05%, which follow the variations of SA mass concentration, while the variations of NCCN at SSs higher than 0.4% were different from the variations of SA mass concentration, especially under high RH conditions. This suggests that the variations of CCN activity at SSs higher than 0.4% are not influenced by SA formation, probably due to the particle size where SA formation occurs is much larger than those dominant on CCN activity for SSs higher than 0.4%. We have added these discussion into section 3.1 as follow:

*"It should be noted that variations of $N_{CCN}$ at 0.07% were similar to those at 0.05%, which followed the variations of SA mass concentration. While at higher SSs, the variations of $N_{CCN}$ differed from those of SA mass concentration, especially under high RH conditions, suggesting different responses of CCN activity towards distinct SA formation processes."*

[Figure]

**Fig S1. Overview of the measurements during the campaign: (a) dots represent wind speed with color indicating wind direction, and black lines represent RH; (b) SPAR under SS of 0.07%; (c) SPAR under SS of 0.2%; (d) blue, green and yellow dots represent NCCN under SS of 0.07% and 0.2%, and mass concentration of NR-PM2.5, respectively; (e) blue and yellow dots represent mass concentration of PM2.5 PA and PM2.5 SA respectively; (f) blue and yellow dots represent ratio between NCCN under SS of 0.07% and mass concentration of NR-PM2.5 and PM2.5 SA, respectively, (g) blue and yellow dots represent ratio between NCCN under SS of 0.2% and mass concentration of NR-PM2.5 and PM2.5 SA, respectively. There were four events with significant enhancements of NCCN during the blue shaded periods.**

[Figure]

**Fig S2.** Diurnal variation of (a) PNSD, (b) SPAR at SS of 0.07%, (c) GF-PDF at 150nm, (d) SPAR at SS of 0.2%, (e) GF-PDF at 100nm and (f) mass fraction of different PM2.5 chemical species during high RH periods before 4th Dec (1) low RH periods after 4th Dec (2) and the four events (3-6), including OA factors: hydrocarbon-like OA (HOA), cooking OA (COA), biomass burning OA (BBOA), coal combustion OA (CCOA), and oxygenated OA (OOA).

And as shown in Figs. S1 and S2, the variations of SPAR and NCCN/PM at SS of 0.07% are similar but lighter, compared with those at SS of 0.05%. While for SS of 0.2%, the difference of SPAR between different periods or events are smaller (Fig. S1), and so did the diurnal variations of SPAR and GF-PDF at particle size of 100nm (Fig. S2). Because CCN activity at SS of 0.2% was strong enough (indicated by SPAR value close to 1) in particle size range where the SA formation dominates, and thus the different SA formations under high or low RH conditions cannot lead to significant variations of CCN activity at SS of 0.2%. In summary, based on CCN measurements in this study, the RH-dependent influence of SA formation on CCN activity can be found obviously at SSs of 0.05% and 0.07%. As the variations of CCN activity at SS of 0.07% were quite similar to those at SS of 0.05, further analysis was only based on CCN activity at SS of 0.05%. We have added a paragraph of these discussions after the first paragraph of section 3.2 (discussing Fig. 2) as follow:

*"Besides SS of 0.05%, variations of SPAR at SSs of 0.07% and 0.2% are also shown in Figs. S1 and S2 in the supplement. And as shown in Figs. S1 and S2, the variations of SPAR and NCCN/PM at SS of 0.07% are similar but lighter, compared with those at SS of 0.05%. While for SS of 0.2%, the difference of SPAR between different periods or events are smaller (Fig. S1), and so did the diurnal variations of SPAR and GF-PDF at particle size of 100nm (Fig. S2). Because CCN activity at SS of 0.2% was strong enough (indicated by SPAR value close to 1) in particle size range where the SA formation dominates, and thus the different SA formations under high or low RH conditions cannot lead to significant variations of CCN activity at SS of 0.2%. In summary, based on CCN measurements in this study, the RH-dependent influence of SA formation on CCN activity can be found obviously at SSs of 0.05% and 0.07%. As the variations of CCN activity at SS of 0.07% were quite similar to those at SS of 0.05, further analysis was only based on CCN activity at SS of 0.05%."*

*3. At many parts of the paper, the MAF (maximum activated fraction) parameter appears (together with a single sigmoid fit) and is used for the fraction of the hygroscopic particles. As I mention later in the detailed comments, this parameterization/fit can only be used in certain cases. You should include a discussion and provide information on how well this fit could be used for your data. And dependent on the SS set in the CCNC, the MAF you present has a different meaning. You only show measurements at SS=0.05%, at this SS and with the highest considered dry diameter of 300nm, this MAF has the meaning of the fraction of the particles having a kappa at least approx. 0.22, far far away from non-hygroscopic. 1/3 AS and 2/3 BC would have such a kappa. Use MAF accordingly, and correctly! And I would need proof that this fitting method can be used for your data at any time during the campaign. For the 0.05% case, it assumes that there are no particle present around the kappa of 0.22, just a population with much higher kappa and a population with significantly lower kappa. Was it the case for the whole measurement period? If there will be other SSs included in the paper following my suggestion, then please check and show what the MAF would mean at that SS, like e.g. at 0.2% and maximum diameter of 300nm, the MAF would be the fraction of the particles having a kappa higher than approx. 0.013. Or a much better choice would be doing such a fit until a constant kappa at different SSs which would mean different maximal fit diameters. That would have a more useful meaning. Like the fraction of the particles having a hygroscopicity below kappa 0.1. That would mean that you have to use the measurements until a higher diameter than 300nm (approx. 390nm) at SS=0.05% which you did not include because of having too much noise. But that problem could be solved following another one of my previous suggestions and using more data and doing some time averaging. You have many choices, choose something which you like, but it is very confusing right now, and this MAF, as calculated now, is not representative for the fraction of the non-hygroscopic particles.*

Response: Thanks for your comment.

We agree that the meaning of MAF can be different regard to the SS and the MAF fitting for SPAR at SS of 0.05% with the highest dry diameter of 300nm cannot represent the non-hygroscopic particles.

Also, it's certainly not true that there are no particle present around the kappa of 0.22, and just a population with much higher kappa and a population with significantly lower kappa during this campaign. In order to represent particle hygroscopicity (kappa) of about 0.1 at SS of 0.05%, SPAR measurement up to about 400nm is needed. We have added the description about this source of uncertainty for SPAR fitting in the methodology:

1. In the second paragraph of section 2.1.2 (description of DMA-CCNC), we have added a sentence as "*In order to characterize the variations of particles with low hygroscopicity of about 0.1, SPAR measurement up to about 400nm is used at 0.05% SS.*"

2. After the first paragraph of section 2.2.4 (description of SPAR fitting), we have added a sentence as "*To be noted, the meaning of MAF can be different regard to the SS, and SPAR measurement up to about 400nm is needed for the MAF fitting for SPAR at SS of 0.05% to represent the particles with kappa higher than 0.1.*"

In addition, we have also improved the fitting of MAF by extending the upper size limit of SPAR to about 400 nm, which corresponds to kappa of about 0.1 at SS of 0.05%, and obtain new fitting parameters, as shown in Fig. R1 below. Compared with original parameters, new MAF and Da are both higher, especially at SSs of 0.05%.

[Figure]

**Fig R1. SPAR and the corresponding fitting parameters of MAF and Da for the original (red) and the expanded particle size ranges (green) at the five measured SSs. The vertical red and green lines indicate the original Da and the new Da, respectively. The vertical black line indicates the particle size of 300nm.**

Furthermore, we have revised the corresponding parts related to SPAR fitting parameters, including Figs. 3, 5, 6 and 7, as shown below. As the temporal variations of SPAR fitting parameters can be expected to be affected little by extending the upper limit of particle size, the diurnal variations of SPAR and its fitting parameters are changed a little bit but the conclusions in Fig. 3 are still valid. In Fig. 5, diurnal variations of the ratios between calculated NCCN and measured NCCN are stronger and the standard deviations are higher. These strong diurnal variations and larger deviations are because both the fitting parameters of MAF and their difference from the campaign averaged MAF become larger. In Figs. 6 and 7, there are difference of MAR_SPAR and the corresponding calculated NCCN (based on MF$_{SA}$ and NF$_{hygro}$) by expanding the size range of SPAR. As the Figs. 6c and 7c show, the calculated NCCN become lower, which is mainly due to the higher values of new Da shown in Fig. R1. Thus, compared with the original results, correlation in Figs. 6b, 6c, 7b and 7c become a little worse (the slopes of the correlation decrease from about 0.99 to 0.89). Nevertheless, as in particle size

range larger than 400nm, the PNSDs are low and the resultant influence on NCCN are small, the conclusions in Figs. 5, 6 and 7 are still valid.

[Figure]

Fig 3. (a) The averages of SPAR curves at SS of 0.05% in three different time periods (blue: 0:00-8:00; green: 8:00-12:00; red: 12:00-16:00) during high (squares with solid line) and low (dots with dashed line) RH events. Diurnal variation of (b) Da and (c) MAF under high (blue) and low (yellow) RH conditions. The blue, green and red shades correspond to with the three periods in (a & d). Error bars indicate the standard deviations of data.

[Figure]

Fig 5. (a) The averaged SPAR during the campaign (green scatters), the corresponding fitting curve (green line) and the averaged fitting parameters (dotted line for Da and dashed line for MAF). The blue and yellow shaded areas represent the variations of SPAR before 4th Dec and after 4th Dec, respectively. The ratio between calculated NCCN and measured NCCN under (b) before 4th Dec and (c) after 4th Dec. Bars represent one standard deviation and colors represent different calculation of SPAR curves: green represent average SPAR during the campaign (AvgSPAR), yellow represent SPAR calculated with average Da and real-time MAF (AvgDa) and blue represent SPAR calculated with average MAF and real-time Da (AvgMAF).

[Figure]

Fig 6. (a) The comparison between calculated $N_{CCN}$ based on kappa derived from bulk particle chemical compositions ($N_{CCN\_chem}$) and measured $N_{CCN}$ at SS of 0.05%. (b) The correlation between MAF and mass fraction of secondary aerosol ($MF_{SA}$). (c) the comparison between calculated $N_{CCN}$ based on SPAR derived from real-time $MFSA$ and average Da ($N_{CCN\_MF}$) and measured $N_{CCN}$. The black dashed lines represent the relative deviation of 30%. (d) the diurnal variations of the ratio between the calculated and measured $N_{CCN}$ during the whole campaign based on different methods (green: $N_{CCN\_chem}$; blue: $N_{CCN}$ based on SPAR derived from averaged $MF_{SA}$ and average $D_a$; yellow: $N_{CCN\_MF}$).

[Figure]

**Fig 7. (a) The comparison between calculated NCCN based on kappa derived from bulk GF at 200 nm (NCCN_GF) and measured NCCN at SS of 0.05%. (b) The correlation between MAF and number fraction of hygroscopic particles (NFhygro, GF>1.2). (c) The comparison between calculated NCCN based on SPAR derived from real-time NFhygro and average Da (NCCN_NF) and measured NCCN. The black dashed lines represent the relative deviation of 30%. (d) the diurnal variations of the ratio between the calculated and measured NCCN during the whole campaign based on different methods (green: NCCN_GF; blue: NCCN calculated based on SPAR derived from averaged NFhygro and average Da; yellow: NCCN_NF).**

*4. Something is strange for me for Figure 5a. How can it be, that the ratio between the calculated and measured N_CCN is systematically below 1? I would expect using the partly or completely averaged SPAR (whichever trace I look at it), that the ratio is scattered around one, but not being always below (like in Figure 5b). For me, this could only happen if e.g. you have a systematic error in the fitting procedure, which always underestimate the measured SPAR, or something else. In my opinion something can not be correct here. Please explain me, if the data is correct, how that can be.*

Response: Thanks for your comment. In Fig. 5a shown in general comment 3 above, the calculated NCCN for AvgSPAR before and after 4$^{th}$ Dec are both on the basis of the averaged SPAR of this

campaign rather than the averaged SPAR before or after 4$^{th}$, because the applicability of the campaign averaged SPAR on the NCCN calculation in the NCP was confirmed in many former study (Deng et al., 2012; Wang et al., 2013; Ma et al., 2016). And the systematically low ratio between the calculated and measured NCCN are due to the generally higher SPAR during high RH period than the averaged SPAR during the campaign, as shown in the Fig. 5. During the low RH periods, SPAR are generally lower than the averaged SPAR of the campaign and the ratio between the calculated and measured NCCN are systematically higher (lasting for the whole night). In addition, it can be confirmed that there is no systematic error in the fitting procedure shown as the fitting curve in Fig. 5a. We have added the explanation into the second paragraph (discussing Fig. 5) as follow:

"*To be noted, $N_{CCN\_AvgSPAR}$ before and after 4$^{th}$ Dec are both on the basis of the averaged SPAR of this campaign (green dots in Fig. 5a) rather than the averaged SPAR before or after 4$^{th}$, because the applicability of the campaign averaged SPAR on the NCCN calculation in the NCP was confirmed in many former studies (Deng et al., 2012; Wang et al., 2013; Ma et al., 2016). During the low RH periods, SPAR are generally lower than the averaged SPAR of the campaign and the ratio between the calculated and measured NCCN are systematically higher (lasting for the whole night).*"

*Detailed comments:*

*1. Line72: "different with those" did you mean here different from those?*

Response: Yes, it should be "*different from those*" and we have revised it accordingly.

*2. Line 86: hydrophobic is a too strong expression here, I guess you mean non-hygroscopic*

Response: Thanks for your suggestion. We have revised this sentence as "*In general, the SA formation can increases the hygroscopicity of particles by adding chemical compounds with lower volatility and higher oxidation state…*"

*3. Line 132: how far was the container form the building of the gas measurements?*

Response: The container was about 80 meters away from the building and there are no taller buildings between them that will block air flow. We have added this information into the manuscript.

*4. Line 139: you mention here the SS and the corrected SS of the CCNC, what is this correction? If it is simply the SS calibration, then you do not need to mention the wrong SS levels, just state the correct ones you determined based on the instrument calibration.*

Response: Thanks for your suggestion. It is the SS calibration and we have revised it accordingly.

*5. Line 154-156: about the inversion and multiple charge correction of the scanning CCNC system: you mention that a multiple charge correction was done and show some references, where details about it can be found. However, I really had to search longer among those papers until I found a method in one of them. Since the main result what you show in this paper is the SPAR, to my opinion the method of inversion/multiple charge correction has to appear a bit more detailed in this paper. And as I understood from the method I found in one of the references (if I found the method you used here), a simple correction only taking the multiple charged particles into account was applied. The width of the DMA transfer function was neglected. Please at least speculate on it, how much error you introduce to your measurement with this assumption.*

Response: Thanks for your comments. In fact, the influence of DMA transfer function has been considered in our inversion method, which is the updated version of Deng et al. (2011) and similar to the inversion method of size distribution of black carbon in our recent study (Zhao et al., 2019). We have added the information about the inversion method into supplements as follow:

"*When the DMA is charged with a negative voltage, those aerosols with a small range of electrical mobility ($Z_P$) can pass through the DMA. When the scan diameter is set as Dpi for the singly charged particles and the respective voltage of DMA is $V_i$ (i = 1, 2, ..., I ), aerosol particles with an electrical mobility of $Z_{p,i}$ (i = 1, 2, ..., I) can pass through the DMA and the observed $N_{CCN}$ by CCN counter can be expressed as:*

$$R_i = \int_0^\infty G(i,x) A(x) n(x) dx \qquad (S1)$$

*where x is the scale parameter with the definition of x = log($D_{pi}$); A(x) is the SPAR of a single particle for scale parameter x; and n(x) = dN/dlog$D_p$ is aerosol PNSD that is the multiple charging corrected results from the measured aerosol PNSD. We define the kernel function G(i,x), which is crucial to the algorithm, as:*

$$G(i,x) = \sum_{\upsilon=1}^\infty \phi(x,\upsilon)\Omega(x,\upsilon,i) \qquad (S2)$$

*where $\phi(x,\upsilon)$ is the probability of particles that are charged with v charges at the scale parameter of x (Wiedensohler, 1988). Transfer function $\Omega(x,\upsilon,i)$ is the probability of particles that can pass through the DMA with v charges at the scale parameter x (Knutson and Whitby, 1975). In this study, the maximum value of $\upsilon$ is 10.*

*The multiple charging corrections can be expressed as computing the A($x_i$\*), in which $x_i$\* is the predetermined scale parameter from the DMA. To get the numerical integration results of Eq. (9), the range of the diameter is [$x_{int,1}$,$x_{int,J}$] and the diameter interval that is 1/50 of the measured diameter is used. For $x_{int,1}$, its mobility is the 50% higher than the mobility of $x_1$\* with single charge. For $x_{int,J}$, its mobility is the 50% higher than the mobility of $x_1$\* with ten charges. Thus, Eq. (S2) can be written*

*as:*

$$R_i = \int_{x_{\text{int},1}}^{x_{\text{int},J}} G(i,x) A(x) n(x) dx = \Delta x_{\text{int}} \sum_{j=1}^{J} \beta_j G(i,x_{\text{int},j}) A(x_{\text{int},j}) n(x_{\text{int},j}) \qquad \text{(S3)}$$

*where* $\beta_j = \begin{cases} 0.5, \ j=1, J \\ 1, \text{otherwise} \end{cases}$, $x_{int,j}$ *is the jth (j=1, 2, ..., J) parameter that locates at the parameter $x_i$ and*

$x_{i+1}$, *and* $A(x_{\text{int},j}), j=1,2,...,J$ *is SPAR at scale parameter $x_{int,j}$, which is expressed as the linear interpolation of the values at the measured diameters:*

$$A(x_{\text{int},j}) = A(x^*_{i(j)}) + P_{i(j)}(x_{\text{int},j} - x^*_{i(j)}) \qquad \text{(S4)}$$

*where $P_i$ is the slope of the linear interpolation result of the five diameters that are nearest to the predetermined scale parameter $x_i$.*

Then by considering

$$H_{ij} = \beta_j \Delta x_{\text{int}} G(i,x_{\text{int},j}) n(x_{\text{int},j}) \qquad \text{(S5)}$$

*the equation (S3) can be rewritten as:*

$$R_i = \sum_{j=1}^{J} H_{ij} A(x_{\text{int},j}) \qquad \text{(S6)}$$

*then*

$$R_i = \sum_{j=1}^{J} H_{ij} \left[ A(x^*_{i(j)}) + P_{i(j)}(x_{\text{int},j} - x^*_{i(j)}) \right]$$

$$= \sum_{j=1}^{J} H_{ij} A(x^*_{i(j)}) + \sum_{j=1}^{J} H_{ij} P_{i(j)} x_{\text{int},j} - \sum_{j=1}^{J} H_{ij} P_{i(j)} x^*_{i(j)}$$

$$= \sum_{k=1}^{I} \left( \sum_{j=1}^{J} H_{ij} \delta(i(j)-k) \right) A(x^*_k)$$

$$+ \sum_{k=1}^{I} \left( \sum_{j=1}^{J} H_{ij} x_{\text{int},j} \delta(i(j)-k) \right) P_k$$

$$- \sum_{k=1}^{I} \left( \sum_{j=1}^{J} H_{ij} \delta(i(j)-k) \right) P_k x^*_k$$

$$= \sum_{k=1}^{I} Q_{ik} A(x^*_k) + \sum_{k=1}^{I} T_{ik} P_k - \sum_{k=1}^{I} Q_{ik} P_k x^*_k \qquad \text{(S7)}$$

*where the Dirac Function is:*

$$\delta(x) = \begin{cases} 0, x \neq 0 \\ 1, x = 0 \end{cases} \qquad \text{(S8)}$$

*thus*

$$Q_{ik} = \sum_{j=1}^{J} H_{ij} \delta\left(i(j) - k\right) \qquad (S9)$$

$$T_{ik} = \sum_{j=1}^{J} H_{ij} x_{\text{int},j} \delta\left(i(j) - k\right) \qquad (S10)$$

*by letting the*

$$S_i = R_i - \sum_{k=1}^{I} T_{ik} P_k + \sum_{k=1}^{I} Q_{ik} P_k x_k^* \qquad (S11)$$

*this equation is then expressed as*

$$S_i = \sum_{k=1}^{I} Q_{ik} A\left(x_k^*\right) \qquad (S12)$$

*or*

$$\mathbf{S=QA} \qquad (S13)$$

*where S and A are I×1 vectors and Q is an I×I matrix. This matrix can be solved by using the non-negative least square method. Finally, the A(x) can be determined and the corresponding size-resolved SPAR that is multiple charging corrected can be calculated."*

*6. Line 170 and 173: "under RH of 90%" please change under to at, under could be also understood as below.*

Response: Thanks for your suggestion. We have revised it accordingly.

*7. Line 179: you mention 4 dry sizes in Line 178 and then 6 sizes in this line. Which one is correct?*

Response: There are 4 dry size and it's a typo in line179. We have revised it accordingly.

*8. Line 209: What function was used for the fit?*

Response: The GF-PDF was not fitted but derived from the measured GF distribution by the TDMAinv algorithm (Gysel et al., 2009). We have revised the description accordingly.

*9. Line 211: "(HGF?)" Typo?*

Response: Yes, it's a typo and we have revised it accordingly.

*10. Line 217: "(Da_hygro)" what is that?*

Response: Da_hygro is the critical diameter for particles with GF_hygro at a certain SS, and the Da_HGF is the critical diameter for particles with average GF. As GF_hygro is higher than the average GF, Da_hygro is smaller than Da_HGF. We have revised this sentence as "*…, the hygroscopicity parameter κ and corresponding critical diameter (Da_{hygro}) under a certain SS for particles with GF_hygro can be calculated. As GF_{hygro} is higher than the average GF, Da_{hygro} is smaller than Da_{HGF}.*".

*11. Line 224: "dominate" change to dominant*

Response: Thanks for your suggestion. We have revised it accordingly.

*12. Line 240: please change "reported in the same. . ." to "reported from the same. . ."*

Response: Thanks for your suggestion. We have revised it accordingly.

*13. Line 243: what are the kappa values you used for the inorganics? The kappa theory is not a perfect parameterization of the water activity, and therefore it is not granted that a kappa you calculate from a HTDMA will be the same as what you get from a CCN measurement. For example, AS has a different kappa at supersaturation and at 90% RH. How did you take this into account? Please comment on it. And at what RH was the relationship for kappa_org determined in the mentioned study?*

Response: Thanks for your comment. We agree that the kappa theory is not perfect and kappa value may vary with RH conditions, even for inorganic compounds. And it's very important to consider the RH conditions when using kappa values of chemical compounds. Numerous studies have focused on the performance of its applications on measurements under different RH conditions (e.g. Liu et al., 2011; Wang et al., 2017). The kappa values for inorganics in Liu et al. (2014) are derived from ISORROPIA II (Fountoukis and Nenes, 2007) under sub-saturated conditions. However, the enhancement of kappa values under super-saturated should be considered in this study, especially for sulfate, and we have been revised to kappa values under super-saturated conditions as follow (Petters et al., 2007):

| Species | $NH_4NO_3$ | NH4HSO4 | (NH4)2SO4 | NH4Cl |
|---------|-----------|---------|-----------|-------|
| κ | 0.67 | 0.7 | 0.61 | 0.93 |

As the mass fraction of sulfate ions during the campaign were generally lower than 20%, the difference of kappa values was generally within 0.02. As for kappa_org, it was determined by the measurement of humidified nephelometer at RH of 85% in Kuang et al., (2020), due to the lack of kappa_org measured under super-saturated conditions. In addition, in this study, we focus on the variations of

kappa values on NCCN calculation derived from different measurement during the SA formation events, rather than a closure of kappa values which will be addressed in an upcoming study. And as for the NCCN calculation, after revised the kappa value of inorganic compounds, there was still large deviation of calculated NCCN from measured NCCN. We have revised the NCCN calculated based on particle chemical compositions in Fig. 6 and added a paragraph about these descriptions in the end of section 2.2.3 as follow:

"*It should be noted that the κ-Köhler theory is not perfect, even for inorganic compounds. Numerous studies have been focusing on the performance of its application on measurements under different RH conditions (Liu et al., 2011; Wang et al., 2017). And $\kappa_{org}$ used in this study was determined by the measurement of humidified nephelometer at RH of 85% in Kuang et al., (2020), due to the lack of $\kappa_{org}$ measured under super-saturated conditions. In this study, we focus on the variations of κ values derived from HTDMA and CCN measurement during the SA formation events, rather than the closure between κ values derived using different techniques, which will be addressed in an upcoming study.*"

*14. Line 248: it is not generally parameterized, often but for sure not generally, please correct*

Response: Thanks for your suggestion. We have revised it accordingly.

*15. Line 249: change hydrophobic to non-hygroscopic, or what kind of hydrophobic particles do you mean? I am not aware of any kind of atmospheric aerosols that are hydrophobic. To my knowledge non-hygroscopic (kappa=0) aerosol particles activate like a completely non-soluble but wettable surface according to the Kelvin-effect. Hydrophobic particles activate at even worse than those, so at a higher SS.*

Response: Thanks for your suggestion. Here we were referring to particles with kappa lower than 0.1, which were thought to be linked with POA in this study. We fully agreed and have revised it throughout the manuscript accordingly.

*16. Next to that, a CCNC can theoretically measure non-hygroscopic activation at any SS, you simply need to get to a high enough particle diameter. So please change the sentence accordingly mentioning, that your used setup, which only goes up to 300nm, was not able to capture the activation.*

Response: Thanks for your suggestion. We have revised this sentence as "*This parameterization assumes aerosols to be an external mixture of apparently hygroscopic particles that can act as CCN and non-hygroscopic particles that cannot be measured by CCNC within the measured particle size range below 400 nm (Rose et al., 2010).*"

*17. Next to it, at your highest SS of 0.8%, non-hygroscopic particles (kappa=0) that have larger dry diameter than 270nm already activate. So, at your highest SS and diameter of 300 nm you should activate the non-hygroscopic particles as well and get an MAF of 1 (assuming now a very narrow DMA transfer function which might not be the case) independent on the fraction of the non-hygroscopic particles.*

Response: Thanks for your comment. For SPAR at SS of 0.8%, it should be 1 at diameter of 300nm. However, a MAF of 1 may lead to an overestimation of the number fraction of hygroscopic particles due to significant difference between SPAR curves and the sigmoidal fitting curves. In our former study on SPAR fitting in the NCP, we found that a fitting parameterization with the combination of two sigmodal fitting curves was needed for SPAR fitting at SSs higher than 0.4% (Tao et al., 2020). However, in this study, we focus on SA formation occurring mainly on accumulation mode particles. Thus at SSs lower than 0.2%, where the non-hygroscopic particles at particle size of 300nm can be CCN-inactive and the fitting of only one sigmodal curves is applied on SPAR curves. The variations of SPAR were prominent in the particle size range smaller than 400 nm rather than larger particle size. And the MAF fitted in this particle size range characterized number fraction of particle with kappa value larger than 0.1 and can be used to indicate the variations of SPAR focused in this study. In addition, due to the very low NCCN in particle size larger than 300 nm, the deviations of NCCN due to the limited range of measured particle size is also very small. We have added a paragraph about these description in the end of section 2.2.4 as follow:

*"For SPAR at SS of 0.8%, MAF should be 1 at 400 nm diameter. However, a MAF of 1 in this case can lead to overestimations of hygroscopic particle number fraction due to the significant difference between SPAR curves and sigmodal fitting curves. In the former study on SPAR fitting curves in the NCP, it was found that a fitting parameterization with the combination of two sigmodal fitting curves was needed for SPAR fitting at SSs higher than 0.4% (Tao et al., 2020). However, in this study, we investigate SA formation on accumulation mode particles and particle CCN activity at SSs below 0.1%, under which condition non-hygroscopic particles smaller than 400 nm are typically CCN-inactive. The MAF fitted in the particle size range below 400 nm was used to indicate the variations of SPAR that was of the main focus here in this work. In addition, due to the very low $N_{CCN}$ in particle size ranges larger than 400 nm, the deviations of $N_{CCN}$ due to the limited range of measured particle size is also very small."*

*16. Line 254: "can represents" do you mean here can represent or represents? The later would not be true, if you have a hygroscopic fraction of the aerosols with not a single kappa but a broader kappa distribution. Please include a discussion on this here.*

Response: Thanks for your comments. In our recent study, based on the investigation of the covariations between SPAR curves and parameterized kappa distribution, it was found that the MAF

can be used to estimate the number fraction of hygroscopic (thus CCN-active) particles at particle size around Da, for kappa distribution of ambient aerosol particles (Jiang et al., 2021). We have revised this sentence as follow:

"*MAF is the asymptote of the measured SPAR curve at large particle sizes. Da is the midpoint activation diameter .... the heterogeneity of CCN hygroscopicity. As reported by Jiang et al. (2021), based on the investigation of the covariations between SPAR curves and parameterized hygroscopicity distribution, it was found that the MAF can be used to estimate the number fraction of hygroscopic (thus CCN-active) particles, for aerosol hygroscopicity distributions generally observed in ambient atmosphere, and thus half MAF can be used represent the number fraction of CCNs to total particles at particle size around Da*"

*15. Line 255: sigma of the error function: does not only include the heterogeneity of the hygroscopicity but also the transfer function of your measurement system, mainly the DMA transfer function.*

Response: Thanks for your comments. As shown in the response to the Detailed comment 3, the influence of transfer function has been considered, thus will not affects the values of sigma here.

*16. Line 257: see my previous comments on "hydrophobic". Kappa<0.1 is not even non-hygroscopic. It would be something like a particle consisting of approx. 17% of AS and 83% of BC. One definitely cannot call this hydrophobic.*

Response: Thanks for your comments. Here we are referring to particles with κ less than 0.1 and we have revised "*nearly hydrophobic particles*" to "*particles whose κ is less than 0.1*".

*17. Line 266: change "is" to "was"*

Response: Thanks for your suggestion. We have revised it accordingly.

*18. Line 290: you show the CCN activation ratio/fraction not activity*

Response: Thanks for your suggestion. We have revised "CCN activity" to "SPAR" here.

*19. Section 3.2: From figure 1 it looks like, that you have a strong diurnal variation of the CCN activity almost every day. Somehow you only show the results of the few selected events. Please show at least an average (and the variation) of all the days for the data you show in Figure 2. And discuss them. It would be also nice to show the diurnal variation of the number size distribution as well.*

Response: Thanks for your suggestion. We have added the average diurnal variations of SPAR (Figs.

2(1b) and (2b) and PNSD in supplement as shown above (Figs. 2(1a) to (6a))and the corresponding discussion into the first paragraph of section 3.2 as mentioned in general comment 1.

*19. Line 301: please correct "hydrophobic"*

Response: Thanks for your suggestion. We revised it accordingly.

*20. Figure 3a: please include the standard deviation of the averages for the SPAR curves as error bars or shading*

Response: Thanks for your suggestion. We have revised Fig. 3a accordingly as shown above.

*21. Figure 3b-c: what are the error bars? The error of the fits, or the standard deviations of the calculated averages, or something else?*

Response: The error bars are the standard deviations of the calculated averages and we have added the description into the caption of Fig. 3b-c.

*22. Line 327-329: I do not understand this sentence*

Response: Thanks for your suggestion. The discussion here is not necessary and may lead to confusion, thus has been removed.

*23. Line 331-33: I do not understand either*

Response: Thanks for your suggestion. As mentioned above, the discussion here is not necessary and may lead to confusion, thus have been removed.

*24. Line 348: "to can be expected" typo*

Response: Thanks for your suggestion and we have deleted "to".

*25. Figure 4b: showing the number of aerosol particles instead of the volume would be much useful, the CCN activity is also measured by the number and not by the volume*

Response: Thanks for your suggestion. The variations of aerosol number concentration shown in Fig. S3 in the supplement have been moved into Fig. 4 and the variations of aerosol volume concentration were kept in Fig.4 to link the variations of SA mass concentration and NCCN.

*26. Line 399-402: Sentence too long, please start a new sentence after "respectively" and reformulate if, it is hardly understandable.*

Response: Thanks for your suggestion. We revised this sentence as "*However, for a unit amount of SA formation, the increase of NCCN was stronger under low RH conditions and weaker under high RH conditions.*"

*27. Line 411: change please "was" to "is"*

Response: Thanks for your suggestion. We have revised it accordingly.

*28. Line 420: do you mean "is calculated based on. . ."?*

Response: No. But this sentence may be confusing and we have revised it as "*The ratio between $N_{CCN}$ calculated based on campaign averaged SPAR ($N_{CCN\_cal}$) to measured $N_{CCN}$ ($N_{CCN\_meas}$) before and after 4$^{th}$ Dec are shown in Fig. 5. SPAR is determined by the variation of Da and MAF, which reflect changes in hygroscopicity and number fraction of hygroscopic particles.*".

*29. Line 420: CCN activity is not a quantity, somehow you use that through the whole paper as it was. Please correct it everywhere. What do you mean by it here? The SPAR? Or some kind of N_CCN? How is the N_CCN_cal exatly defined? Or is that the calculated N_CCN? Please rewrite this whole sentence and explain how you exactly calculated the CCN prediction.*

Response: Thanks for your suggestion and we have corrected the use of "CCN activity" throughout the manuscript. It referred to as the SPAR here and we have revised this sentence as mentioned in the previous comment.

*30. Line 426: "as to" -> "to as"*

Response: Thanks. We have revised it accordingly.

*31. Line 439-442: For me it would be strange if using a completely different instrument for a kappa measurement from bulk chemistry assuming internally mixed aerosols would improve the N_CCN prediction compared to the prediction based on the averaged SPAR. Please do not introduce this prediction method as an improvement.*

Response: Thanks for your suggestion and we have revised this description as "*provide calculation of $N_{CCN}$ combining with PNSD measurement with smaller deviations*".

*32. Line 440: please include the exact definition of the number fraction of hygroscopic particles!*

Response: Thanks for your suggestion. The number fraction of hygroscopic particles is defined in the equation (3) in section 2.2.2, and we have revised it as "*Number Fraction of hygroscopic particles (GF(90%, 200nm)>1.22, NF$_{hygro}$)*".

*33. Line 453: calling R^2=0.59 a "strong correlation" is maybe a little bit too strong.*

Response: Thanks for your suggestion and we have revised "*strong correlation*" to "*positive correlation*".

*34. Figure 6: Please show the calculated vs. measured N_CCN for the methods you used for Figure 5 as well to have a comparison.*

Response: Thanks for your suggestion. We have added the diurnal variations of the ratios between the calculated and measured N_CCN into Fig. 6 as shown in general comment 3 above, and the corresponding discussions into the manuscript as: "*Additionally, the diurnal variations of $N_{CCN\_cal}$/$N_{CCN\_meas}$ ratio based on different methods of $N_{CCN}$ calculation during the whole campaign were shown in Fig. 6(d). It can be found that by considering the real-time MF$_{SA}$, the deviation of calculated $N_{CCN}$ (Real-time MF in Fig. 6d) can be reduced throughout the day, compared with $N_{CCN\_Chem}$ (Real-time Chem in Fig. 6d). Meanwhile, if an averaged MF$_{SA}$ is used to estimate SPAR and $N_{CCN}$, the deviations of calculated $N_{CCN}$ (Averaged MF in Fig. 6d) can be reduced as well, but demonstrated a much stronger diurnal variations than the deviation of $N_{CCN\_MF}$.*"

We have also revised Fig. 7 in a similar way and added the corresponding discussions as "*The diurnal variations of the $N_{CCN\_cal}$/$N_{CCN\_meas}$ ratio based on different methods of $N_{CCN}$ calculation during the whole campaign were shown in Fig. 7(d). It can be found that by considering the real-time NF$_{hygro}$, the deviation of $N_{CCN\_NF}$ (Real-time NF in Fig. 7d) can be reduced mainly during nighttime, compared with $N_{CCN\_GF}$ (Real-time GF in Fig. 7d). Meanwhile, if an averaged NF$_{hygro}$ is used to estimate SPAR and $N_{CCN}$, the deviations of calculated $N_{CCN}$ (Averaged NF in Fig. 7d) can be reduced during nighttime as well, but demonstrated a much stronger diurnal variations than the deviation of $N_{CCN\_NF}$.*".

*35. Line 459-472: you could not only use the bulk HTDMA hygroscopicity but the complete GF-PDF for the N_CCN estimation considering the mixing state of the aerosols as well. For sure, that would improve the calculation as well.*

Response: Thanks for your suggestion. We have calculated NCCN based on GF-PDF at 200nm by assuming constant GF-PDF in all particle size range and compared with measured NCCN in Fig S5 shown below. This simplified method to deal with GF-PDF is mainly to due to limited measured

particle sizes of HTDMA, but still applicable for NCCN calculation because CCN at SS of 0.05% mainly distribute in particle size range from 200nm to 300nm, where the difference between GF-PDF at 200nm and 250nm was generally small, as shown in Figs. 2 and S2. In addition, as Fig. S5 shown, the calculated NCCN based on GF-PDF agree well with measured NCCN as the mixing state of aerosol is considered and also support our results that the mixing state of aerosol is important for NCCN calculation. In addition, compared with GF-PDF, calculation with NF_hygro is much easier in application, thus is focused in study.

[Figure]

**Fig. S4. Comparison between the calculated NCCN based on GF-PDF and the measured NCCN.**

We have added this figure into the supplements and added the discussion about GF-PDF into the manuscript as follow:

"*If GF-PDF were directly used to calculate $N_{CCN}$, $N_{CCN\_cal}$ would agree well with measured $N_{CCN}$ (Fig. S5), because in this way the mixing state of aerosol would have been accounted for. However, compared to the approach using GF-PDF, $NF_{hygro}$ is easier to apply in $N_{CCN}$ calculation and can yield similar accuracies.*"

*36. Line 473-478: If you want to show the importance of the changing MAF in the N_CCN prediction then you do not need all these calculations using the HTDMA and the AMS and the MAF prediction based on a whatever measured parameter of these instruments. Just simply show the calculated N_CCN (averaged MAF) vs the measured N_CCN (MAF as it was measued) as you calculated for the orange line in Figure 5. And as it looks like from Figure 5 you would not have an average error higher than 10% using the averaged MAF, so I am really not convinced about your summary statement. It might be important to take an average MAF different from 1 into account, but most probably not its time variation.*

Response: Thanks for your comment. It's true that by considering an averaged MAF a good prediction of NCCN can still be achieved. However, in practice, the time-dependent MAF from measurement of

either HTDMA or AMS are needed to obtain an averaged MAF. In addition, as shown in the corrected Fig. 5 shown above, the deviations of calculated NCCN based on the averaged MAF can be large under low RH conditions, and the use of time-dependent MAF can eliminate these deviations. Thus, in this study, the averaged MAF is not discussed and the application of the time-dependent MAF is highlighted. We have added these discussion into the manuscript as follow:

"*Although a good prediction of $N_{CCN}$ was achieved by applying an averaged MAF (Figs. 5, 6d and 7d), in practice, this would still require CCN measurements or HTDMA/chemical composition measurements as proxies. Additionally, deviations of $N_{CCN\_cal}$ based on the averaged MAF can be large under low RH conditions (Fig. 5c), while time-dependent MAF can eliminate a great part of these deviations. Thus, by replacing MAF with real-time $MF_{SA}$ or $NF_{hygro}$ when deriving SPAR curve, the calculation of $N_{CCN}$ can be significantly improved.*"

Reference:

Wang, Z.B., Hu, M., Sun, J.Y., Wu, Z.J., Yue, D.L., Shen, X.J., Zhang, Y.M., Pei, X.Y., Cheng, Y.F., Wiedensohler, A., 2013. Characteristics of regional new particle for- mation in urban and regional background environments in the North China Plain. Atmos. Chem. Phys. 13, 12495–12506.

Deng, Z.Z., Zhao, C.S., Ma, N., Ran, L., Zhou, G.Q., Lu, D.R., Zhou, X.J., 2013. An ex- amination of parameterizations for the CCN number concentration based on in situ measurements of aerosol activation properties in the North China Plain. Atmos. Chem. Phys. 13, 6227–6237.

Ma, N., Zhao, C., Tao, J., Wu, Z., Kecorius, S., Wang, Z., Größ, J., Liu, H., Bian, Y., Kuang, Y., Teich, M., Spindler, G., Müller, K., van Pinxteren, D., Herrmann, H., Hu, M., Wiedensohler, A., 2016. Variation of CCN activity during new particle formation events in the North China Plain. Atmos. Chem. Phys. 16, 8593–8607.

Deng, Z. Z., Zhao, C. S., Ma, N., Liu, P. F., Ran, L., Xu, W. Y., Chen, J., Liang, Z., Liang, S., Huang, M. Y., Ma, X. C., Zhang, Q., Quan, J. N., Yan, P., Henning, S., Mildenberger, K., Sommerhage, E., Schäfer, M., Stratmann, F. and Wiedensohler, A.: Size-resolved and bulk activation properties of aerosols in the North China Plain, Atmos. Chem. Phys., 11(8), 3835–3846, doi:10.5194/acp-11-3835-2011, 2011.

Zhao, G., Tao, J., Kuang, Y., Shen, C., Yu, Y., and Zhao, C.: Role of black carbon mass size distribution in the direct aerosol radiative forcing, Atmos. Chem. Phys., 19, 13175–13188, https://doi.org/10.5194/acp-19-13175-2019, 2019.

Wang, Z., Cheng, Y., Ma, N., Mikhailov, E., Pöschl, U. and Su, H.: Dependence of the hygroscopicity parameter κ on particle size, humidity and solute concentration: implications for laboratory experiments, field measurements and model studies, Atmos. Chem. Phys. Discuss., 2017, 1–33, https://doi.org/10.5194/acp-2017-253, 2017.

Liu, P. F., Zhao, C. S., Göbel, T., Hallbauer, E., Nowak, A., Ran, L., Xu, W. Y., Deng, Z. Z., Ma, N., Mildenberger, K., Henning, S., Stratmann, F., and Wiedensohler, A.: Hygroscopic properties of aerosol particles at high relative humidity and their diurnal variations in the North China Plain, Atmos. Chem. Phys., 11, 3479–3494, https://doi.org/10.5194/acp-11-3479-2011, 2011.

Kuang, Y., He, Y., Xu, W., Zhao, P., Cheng, Y., Zhao, G., Tao, J., Ma, N., Su, H., Zhang, Y., Sun, J., Cheng, P., Yang, W., Zhang, S., Wu, C., Sun, Y., and Zhao, C.: Distinct diurnal variation in organic aerosol hygroscopicity and its relationship with oxygenated organic aerosol, Atmos. Chem. Phys., 20, 865–880, https://doi.org/10.5194/acp-20-865-2020, 2020.

Wiedensohler, A.: An approximation of the bipolar charge distribu- tion for particles in the submicron size range, J. Aerosol Sci., 19, 387–389, 1988.

Knutson, E. O. and Whitby, K. T.: Aerosol classification by electric mobility: apparatus, theory, and applications, Jo. Aerosol Sci., 6, 443–451, 1975.

Tao, J., Kuang, Y., Ma, N., Zheng, Y., Wiedensohler, A., 2020. An improved parameterization scheme for size-resolved particle activation ratio and its application on comparison study of particle hygroscopicity measurements between HTDMA and DMA-CCNC. Atmos. Environ. 226, 117403. https://doi.org/10.1016/j. atmosenv.2020.117403.

Jiang, X., Tao, J., Kuang, Y., Hong, J. and Ma, N.: Mathematical derivation and physical interpretation of particle size-resolved activation ratio based on particle hygroscopicity distribution: Application on global characterization of CCN activity, Atmospheric Environment, 246, 118137, https://doi.org/10.1016/j.atmosenv.2020.118137, 2021.

Gysel, M., McFiggans, G. B. and Coe, H.: Inversion of tandem differential mobility analyser (TDMA) measurements, Journal of Aerosol Science, 40(2), 134–151, doi:10.1016/j.jaerosci.2008.07.013, 2009.

Fountoukis, C. and Nenes, A.: ISORROPIA II: a computationally efficient thermodynamic equilibrium model for K -Ca2+-Mg2+-NH +-Na+-SO 2-NO −-Cl−-H O aerosols, Atmos. Chem. Phys., 7, 4639–4659, doi:10.5194/acp-7-4639-2007, 2007.

Reviewer #2:

*In the manuscript "Secondary aerosol formation alters CCN activity in the North China Plain", the authors conducted a field study in North China Plain and investigated the influence of second aerosol (SA) formation on CCN activity and on the calculated CCN number concentrations derived from particle number size distribution (PNSD). The CCN activity at 0.05% supersaturation (SS) was discussed. The authors focused on CCN activation at low SS where mainly accumulation mode particles act as CCN and thus on cases of SA in the presence of accumulation mode particles. They found that at two different RH, SA formation had different influence on CCN activity of aerosols. At high RH (minimum RH>50%), SA mass mostly added to larger particles (>300 nm), which resulted in weaker enhancement of CCN activity for per SA mass added as these larger particles were already CCN-active before SA formation. At low RH (minimum RH<30%), SA mass mostly added to smaller particles (<300 nm), which resulted in stronger enhancement of CCN activity for per SA mass added as smaller particles were not CCN-active before SA formation. In addition, they parameterized maximum activation fraction (MAF) using the correlation of MAF with hygroscopic particle number fraction or with mass fraction of SA. The calculated CCN concentrations derived from PNSD using parameterized MAF, campaign average activation diameter and width of activation curve matched better with measured ones compared with using PNSD and kappa from either chemical composition or hygroscopic growth.*

*How aerosol formation and growth affect CCN activity is an important question. The manuscript provides valuable case study on how secondary aerosol formation influence CCN activity for low stratus clouds and fogs. This study carried out comprehensive measurement of aerosol related to CCN activity/hygroscopicity. The findings are interesting. However, I have some comments about the manuscript to address before it is considered for publishing in ACP.*

Response: Thanks for your comments. Suggestions and comments are addressed point-by-point and corresponding responses are listed below.

*Major comments:*

*1. The manuscript only discussed the results at 0.05% SS. How do the findings depend on SS? What about the results at other SS such as 0.1% and 0.2% SS, which is also typical for low stratus clouds? In addition, I suggest explicitly specifying SS when CCN activity or CCN concentration is discussed.*

Response: Thanks for your suggestion. We have added the variations of CCN number concentration at the five measured SSs into Fig. 1, the variations of SPAR and the ratios between CCN number concentration and PM2.5 at SS of 0.07% and 0.2% in Fig. S1 and the diurnal variations of SPAR at SS of 0.07% and 0.2% in Fig. S2, as follow:

[Figure]

**Fig 1. Overview of the measurements during the campaign: (a) dots represent wind speed with color indicating wind direction, and black lines represent RH; (b) SPAR under SS of 0.05%; (c) blue, green and yellow dots represent NCCN under SS of 0.05% and 0.07%, and mass concentration of NR-PM2.5, respectively; (d) blue, green and yellow dots represent NCCN under SS of 0.2%, 0.44% and 0.81%, respectively; (e) blue and yellow dots represent mass concentration of PM2.5 PA and PM2.5 SA respectively; (f) blue and yellow dots represent ratio between NCCN under SS of 0.05% and mass concentration of NR-PM2.5 and PM2.5 SA, respectively. There were four events with significant enhancements of NCCN during the blue shaded periods.**

As the Fig. 1 shows, the variations of NCCN at 0.07% were similar to those at 0.05%, which follow the variations of SA mass concentration, while the variations of NCCN at higher SSs including 0.4% were different from the variations of SA mass concentration, especially under high RH conditions, suggesting different responses to SA formation. We have added these discussions into the first paragraph of section 3.1 as follow:

"*It should be noted that variations of $N_{CCN}$ at 0.07% were similar to those at 0.05%, which followed the variations of SA mass concentration. While at higher SSs, the variations of $N_{CCN}$ differed from those of SA mass concentration, especially under high RH conditions, suggesting different responses of CCN activity towards distinct SA formation processes.*"

[Figure]

**Fig S1. Overview of the measurements during the campaign: (a) dots represent wind speed with color indicating wind direction, and black lines represent RH; (b) SPAR under SS of 0.07%; (c) SPAR under SS of 0.2%; (d) blue, green and yellow dots represent NCCN under SS of 0.07% and 0.2%, and mass concentration of NR-PM2.5, respectively; (e) blue and yellow dots represent mass concentration of PM2.5 PA and PM2.5 SA respectively; (f) blue and yellow dots represent ratio between NCCN at SS of 0.07% and mass concentration of NR-PM2.5 and PM2.5 SA, respectively. (g) blue and yellow dots represent ratio between NCCN at SS of 0.2% and mass concentration of NR-PM2.5 and PM2.5 SA, respectively. There were four events with significant enhancements of NCCN during the blue shaded periods.**

[Figure]

**Fig S2.** Diurnal variation of (a) PNSD, (b) SPAR at SS of 0.07%, (c) GF-PDF at 150nm, (d) SPAR at SS of 0.2%, (e) GF-PDF at 100nm and (f) mass fraction of different PM2.5 chemical species during high RH periods before 4th Dec (1) low RH periods after 4th Dec (2) and the four events (3-6), including OA factors: hydrocarbon-like OA (HOA), cooking OA (COA), biomass burning OA (BBOA), coal combustion OA (CCOA), and oxygenated OA (OOA).

As shown in Figs. S1 and S2, the variations of SPAR and NCCN/PM at SS of 0.07% are similar but lighter, compared with those at SS of 0.05%. For SS of 0.2%, the difference of SPAR between different periods or events are smaller (Fig. S1), and so did the diurnal variations of SPAR and GF-PDF at particle size of 100 nm (Fig. S2). While for SS of 0.2%, the difference of SPAR between different periods or events are smaller (Fig. S1), and so did the diurnal variations of SPAR and GF-PDF at particle size of 100nm (Fig. S2). Because CCN activity at SS of 0.2% was strong enough (indicated by SPAR value close to 1) in particle size range where the SA formation dominates, and thus the different SA formations under high or low RH conditions cannot lead to significant variations of CCN activity at SS of 0.2%. In summary, based on CCN measurements in this study, the RH-dependent influence of SA formation on CCN activity can be found obviously at SSs of 0.05% and 0.07%. As the variations of CCN activity at SS of 0.07% were quite similar to those at SS of 0.05,

further analysis was only based on CCN activity at SS of 0.05%. We have added a paragraph of these discussions after the first paragraph of section 3.2 (discussing Fig. 2) as follow:

"*Besides SS of 0.05%, variations of SPAR at SSs of 0.07% and 0.2% are also shown in Figs. S1 and S2 in the supplement. And as shown in Figs. S1 and S2, the variations of SPAR and NCCN/PM at SS of 0.07% are similar but lighter, compared with those at SS of 0.05%. While for SS of 0.2%, the difference of SPAR between different periods or events are smaller (Fig. S1), and so did the diurnal variations of SPAR and GF-PDF at particle size of 100nm (Fig. S2). Because CCN activity at SS of 0.2% was strong enough (indicated by SPAR value close to 1) in particle size range where the SA formation dominates, and thus the different SA formations under high or low RH conditions cannot lead to significant variations of CCN activity at SS of 0.2%. In summary, based on CCN measurements in this study, the RH-dependent influence of SA formation on CCN activity can be found obviously at SSs of 0.05% and 0.07%. As the variations of CCN activity at SS of 0.07% were quite similar to those at SS of 0.05, further analysis was only based on CCN activity at SS of 0.05%.*"

In addition, we have also added the specification of the SS where CCN activity and CCN number concentration are discussed in the manuscript.

*2. I was somewhat surprised to notice that MAF only reached 0.4-0.6 in Fig. 3. Why were the data larger than 300 nm excluded (L148)? Did the activation fraction reach around one at larger sizes? If one fit Eq.7 to e.g. blue curves in Fig. 3a only till 300 nm, the Da derived at half MAF might be incorrect.*

Response: Thanks for your comments. The low value of MAF was mainly due to the limited range of measured particle size and the large fraction of POA with low hygroscopicity which can be seen in the measurement of particle chemical compositions. The reason of excluding the data for larger than 300 nm is that there is higher noise in CCN measurement due to the very low particle number concentration in this size range. To evaluate the influence of the size cut-off, we have expanded the upper size limit of SPAR to about 400 nm and obtained new fitting parameters as shown in Fig. R1. Compared with the original parameters, new MAF and Da are both higher, especially at SSs of 0.05%. We have also applied these new values of fitting parameters into our study and revised the manuscript accordingly, as described below.

[Figure]

**Fig. R1. SPAR and the corresponding fitting parameters of MAF and Da for the original (red) and the expanded particle size ranges (green) at the five measured SSs. The vertical red and green lines indicate the original Da and the new Da, respectively. The vertical black line indicates the particle size of 300nm.**

Furthermore, we have revised the corresponding parts related to SPAR fitting parameters, including Figs. 3, 5, 6 and 7 (shown below). In the particle size range larger than 300 nm, the SPAR is still lower than 1 at SS of 0.05% (Fig. 3). This is because for particle size of ~ 390 nm, kappa value higher than 0.1 is needed for CCN activation at SS of 0.05%. As the temporal variations of SPAR fitting parameters stay the same, the conclusions based on Fig. 3 are still valid. In the updated Fig. 5, diurnal variations of the ratios between calculated NCCN and measured NCCN are stronger and the standard deviations are higher. These strong diurnal variations and larger deviations are because both the fitting parameters of MAF and their difference from the campaign averaged MAF become larger. In Figs. 6 and 7, there are difference of MAR_SPAR and the corresponding calculated NCCN (based on MF$_{SA}$ and NF$_{hygro}$) by expanding the size range of SPAR. As the Figs. 6c and 7c show, the calculated NCCN become lower, which is mainly due to the higher values of new Da shown in Fig. R1. Thus, compared with the original results, correlations in Figs. 6b, 6c, 7b and 7c are nearly the same except

that the slopes decrease by about 0.1. Nevertheless, as in particle size range larger than 400nm, the PNSDs are low and the resultant influence on NCCN are small, the conclusions in Figs. 5, 6 and 7 are still valid.

[Figure]

**Fig 3. (a) The averages of SPAR curves at SS of 0.05% in three different time periods (blue: 0:00-8:00; green: 8:00-12:00; red: 12:00-16:00) during high (squares with solid line) and low (dots with dashed line) RH events. Diurnal variation of (b) Da and (c) MAF under high (blue) and low (yellow) RH conditions. The blue, green and red shades correspond to with the three periods in (a & d). Error bars indicate the standard deviations of data.**

[Figure]

**Fig 5. (a) The averaged SPAR during the campaign (green scatters), the corresponding fitting curve (green line) and the averaged fitting parameters (dotted line for Da and dashed line for MAF). The blue and yellow shaded areas represent the variations of SPAR before 12-04 and after 12-04, respectively. The ratio between calculated NCCN and measured NCCN under (b) before 12-04 and (c) after 12-04. Bars represent one standard deviation and colors represent different calculation of SPAR curves: green represent average SPAR during the campaign (AvgSPAR), yellow represent SPAR calculated with average Da and real-time MAF (AvgDa) and blue represent SPAR calculated with average MAF and real-time Da (AvgMAF).**

[Figure]

**Fig 6. (a) The comparison between calculated $N_{CCN}$ based on kappa derived from bulk particle chemical compositions ($N_{CCN\_chem}$) and measured $N_{CCN}$ at SS of 0.05%. (b) The correlation between MAF and mass fraction of secondary aerosol ($MF_{SA}$). (c) the comparison between calculated $N_{CCN}$ based on SPAR derived from real-time $MFSA$ and average Da ($N_{CCN\_MF}$) and measured $N_{CCN}$. The black dashed lines represent the relative deviation of 30%. (d) the diurnal variations of the ratio between the calculated and measured $N_{CCN}$ during the whole campaign based on different methods (green: $N_{CCN\_chem}$; blue: $N_{CCN}$ calculated based on SPAR derived from averaged $MF_{SA}$ and average $D_a$; yellow: $N_{CCN\_MF}$).**

[Figure]

**Fig 7. (a) The comparison between calculated NCCN based on kappa derived from bulk GF at 200 nm (NCCN_GF) and measured NCCN at SS of 0.05%. (b) The correlation between MAF and number fraction of hygroscopic particles (NFhygro, GF>1.2). (c) The comparison between calculated NCCN based on SPAR derived from real-time NFhygro and average Da (NCCN_NF) and measured NCCN. The black dashed lines represent the relative deviation of 30%. (d) the diurnal variations of the ratio between the calculated and measured NCCN during the whole campaign based on different methods (green: NCCN_GF; blue: NCCN calculated based on SPAR derived from averaged NFhygro and average Da; yellow: NCCN_NF).**

*3. The authors reported two cases at high RH and only one case at low RH. It would be helpful to discuss how general these conclusions are regarding the influence of SA on CCN activity. The authors seem to indicate that RH is the dominant factor. What about other conditions? For example, how would the size and chemical composition of existing particle affect the conclusion here?*

Response: Thanks for your comments.

We agree that it's important to convince the different responses of CCN activity to different SA formations. In the revised discussions of Fig. 2, the averaged variations of CCN activity during high or low RH conditions are analyzed in front of the analyses of specific events. And as show in revised

Figs. 2(1a-1d) and 2(2a-2d) (shown below), different variations of SPAR to SA formations can be found during periods with different RH conditions. The variations of SPAR, GF-PDF and mass fraction of particle chemical compositions during the periods of high (or low) RH conditions were similar but less significant, as those during high-RH events 1 and 2 (or low-RH events 3 and 4). The four specific events (adding the 14[th] Dec as an events under low RH conditions) with significant variations of CCN activity during SA formation are analyzed as examples (Figs. 2(3x) to 2(6x)). In addition, we have also revised Figs. 1, 3 and 4 accordingly, and the corresponding results in these figures are still valid. We have added corresponding discussion into the first paragraph of section 3.2 as follow:

"*The diurnal averages of PNSD, SPAR at SS of 0.05%, GF-PDF for 200 nm particle and mass fraction of particle chemical compositions during high RH periods before 4[th] Dec, low RH periods after 4[th] Dec and the four events are shown in Fig. 2, respectively. To be noted, ... CCN behavior. As can be seen in Figs. 2 (1b) and (2b), different variations of SPAR due to SA formations can be found during the periods with different RH conditions. The average diurnal variations of these parameters for the entire high RH stage and low RH stage as shown in Figs. 2 (1a-1d) and (2a-2d) revealed similar but more smoothed variations as in the four selected events. The four events are discussed and intercompared in the following to magnify the differences under distinct RH conditions.*"

Furthermore, in this study, the main point is that different SA formations during high RH and low RH environments are responsible for the variations of CCN activity. The "high (or low) RH events" is used to refer to the SA formation events under high (or low) RH conditions for convivence. As reported by Kuang et al., (2020), SA formation mechanisms and the corresponding influence on PNSD and particle chemical compositions are different during periods with different RH conditions. Thus, we investigated the variations of CCN activity measured during the same campaign and found that different SA formations can largely influence CCN activity due to variations of PNSD and particle chemical compositions. The misleading statements in the manuscript have been revised accordingly:

1. After the first sentence in Sec. 3.2. (discussing the Fig. 2), a description has been added as "*To be noted, the "high (or low) RH events" is used to refer to the SA formation events under high (or low) RH conditions for convivence, and it doesn't mean that RH caused variations of CCN behavior.*"

2. The first sentence of the second paragraph in Sec. 3.2 (discussing the Fig. 3a) has been revised as "*In Figs. 3a,d, detailed comparison of particle CCN activity during SA formation events of NCCN enhancements under different RH conditions are shown as the variations of SPAR curves.*"

3. The second sentence of the second paragraph in Sec. 3.3 (discussing the Fig. 5) has been revised as "*In former discussions, CCN activity (indicated by SPAR) revealed significant diurnal variations during this campaign, which were different during SA formations under distinct RH conditions.*"

4. The first sentence of the last paragraph in Sec. 3.3 (the summary of this section) has been revised as "*In summary, MAF exhibited strong diurnal variation that varied under different RH conditions due to different SA formation mechanisms, which ...*"

[Figure]

**Fig 2. Diurnal variation of (a) PNSD, (b) SPAR at SS of 0.05%, (c) GF-PDF at 200nm and (d) mass fraction of different PM2.5 chemical species during high RH periods before 4th Dec (1), low RH periods after 4th Dec (2) and the four events (3-6), including OA factors: hydrocarbon-like OA (HOA), cooking OA (COA), biomass burning OA (BBOA), coal combustion OA (CCOA), and oxygenated OA (OOA).**

*4. I had some difficult time reading the manuscript. I suggest the authors streamlining the writing substantially. Additionally, there are numerous language problems. For example, in many cases, a space was missing before a unit. More specific problems are listed below.*

Response: Thanks for your suggestions. We have streamlining the writing substantially and fixed language problems. These specific problems are addressed point-by-point below.

*Specific comments*

*1. L254, it is half MAF that can represent the number fraction of CCNs to total particles at particle size around Da. Also "represents" should be "represent".*

Response: Thanks for your comment. We have revised them accordingly.

*2. L264, how are PA and SA characterized?*

Response: During the campaign, PM2.5 PA were generally lower than 100 μg/cm$^3$ under both high and low RH periods. Meanwhile, PM2.5 SA can approach about 400 μg/cm$^3$, especially during the strong SA formation events under high RH conditions, but can be lower than 100 μg/cm$^3$ under low

RH conditions. We have added these information into the manuscript as "*During the campaign, PM$_{2.5}$ PA were generally lower than 100 μg m$^{-3}$ under both high and low RH periods. Meanwhile, PM$_{2.5}$ SA can approach about 400 μg m$^{-3}$, especially during the strong SA formation events under high RH conditions, but can be lower than 100 μg m$^{-3}$ under low RH conditions.*".

*3. L276, how are the time ranges of these events defined? By PM2.5 concentration?*

Response: These events were chosen based on not only the RH but also the enhancement of SA. During event 3, the wind speed was generally low, the RH followed a general diurnal variations and SA mass grew steadily and continuously. Thus the interference of the variations of air mass and short-term local emissions can be eliminated and the influence of SA formation can be highlighted. The time window was fixed to two days for the convenience of intercomparing. We have added these descriptions into the manuscript as follow:

"*These events were selected based on the similarity of PM$_{2.5}$ concentration and evolution, while the time window was fixed to two days for the convenience of intercomparing. In addition, during these events, the wind speed was generally low, the RH followed a general diurnal variations and SA mass grew steadily and continuously. Thus the interference of the variations of air mass and short-term local emissions can be eliminated and the influence of SA formation can be highlighted.*"

*4. L283-286, for me it is hard to tell from the data that the ratios were really lower after 4th Dec. Nor can I discern the "decreasing trends".*

Response: Thanks for your comments. It should be during the high RH events before 4th Dec when there were lower ratios and decreasing trends, and we have revised this sentence as "*However, the ratios between N$_{CCN}$ and mass concentration of PM$_{2.5}$ SA or NR-PM$_{2.5}$, were lower during the high RH period and demonstrated strong decreases, especially in Event 1 and 2.*"

*5. L294, by "the increase of hygroscopic particles", do you mean number or mass concentration?*

Response: It refers to the number concentration and we have revised it accordingly.

*6. L304, is this statement necessarily true?*

Response: Thanks for your comments. We have revised this sentence as "*larger variation in CCN activity was observed in Events 3 and 4*"

*7. L311, by which metric do you define "CCN activity"? Do you refer to activation fraction?*

Response: It refer to the size-resolved activation fraction rather than bulk activation fraction. The bulk activation fraction is determined by not only size-resolved activation fraction but also PNSD. Here we focus on particle hygroscopicity which is linked with particle chemical compositions and indicated by size-resolved activation fraction. We have revised the sentence as: "*Same as demonstrated in Fig. 2, SPAR was generally higher and thus particle CCN activity were generally stronger in high RH events than those in low RH events.*"

*8. L313, "the enhancement of particle CCN activity was stronger in low RH events", which metric or data is this statement based on?*

Response: As mentioned in comment 7 above, "*CCN activity*" here refer to the SPAR as well. As shown in Fig. 3a, the difference between SPAR in high and low RH events at 300 nm decreased from 0.2 to 0.1 during the SA formations, indicating for a stronger enhancement in low RH events, probably due to both the stronger increase of SA mass fraction and the higher nighttime PA mass fraction (Fig. 2(e)). We have revised "*CCN activity*" to "*SPAR*" and added these description into the manuscript.

*9. L319, it is not obvious to tell if there is "the increases of Da".*

Response: Thanks for your suggestions. The increase of Da is not significant and we have revised the sentence as "*This can be attributed to the strong increase of MAF and the slight increase of Da, which indicates significant increasing number fraction yet slightly weakening hygroscopicity of hygroscopic particles, respectively.*"

*10. L321, again "the enhancement of CCN activity was lighter", what metric or data is this statement based on?*

Response: As mentioned in comment 7 above, "*CCN activity*" here refer to the SPAR as well. The enhancement of SPAR here refers to the description in Line 313 as shown in Fig. 3 (a). We have improved the description as "*Overall, the enhancement of SPAR was weaker but occurred at a broader particle size range in high RH events than in low RH events, as shown in Fig. 3a.*"

*11. L325-327, "unchanged CCN activity at low RH conditions", how is this statement drawn? Is this finding also valid for other SS?*

Response: As mentioned in comment 7 above, "*CCN activity*" here refer to the SPAR as well. The discussion here is not necessary but may lead to confusion, thus have been removed.

*12. L339, "relatively smaller variations of particle density", this needs support from data or literature.*

Response: Thanks for your suggestion. Based on measurements in the North China Plain, the variations of the accumulation mode particle density ranges from 1.2 to 1.8, whose relative variations are within 20% (Hu et al., 2012; Zhao et al., 2019). We have added this information into the manuscript.

*13. L365, "decreased continuously", it seems not to be a continuous decrease.*

Response: Thanks for your comments. There is increase of NCCN(<300nm)/NR at early times of the SA formation before the decrease of NCCN(<300nm)/NR. So we have deleted "*continuously*".

*14. L445-447, it is not surprising that the correlation of NCCN_chem with NCCN_meas was not good as kappa was only derived from chemical composition of the bulk aerosol, which is highly biased to larger particles.*

Response: Thanks for your comments. We agree that there may be significant deviations in the kappa estimated based on chemical composition of the bulk aerosol, which leads to significant deviations of NCCN prediction. However, in practice, the measurements of size-resolved particle chemical compositions are not common, and chemical composition of the bulk aerosol is still commonly applied in CCN studies (Zhang et al., 2014; Zhang et al., 2016; Che et al., 2017; Cai et al., 2018), especially when particle hygroscopicity measurements were in lack. In addition, we focus on the comparison between the different methods of applying the bulk aerosol chemical composition on NCCN calculation to provide a better method applicable for NCCN calculation on the NCP. We have added these descriptions into the fourth paragraph of section 3.2 as follow:

"*Although there can be significant deviations for $\kappa$ of accumulation mode particles derived from chemical composition of the bulk aerosol, which leads to significant deviations of $N_{CCN}$ prediction. However, in practice, the measurements of chemical compositions of accumulation mode particles are not common, and chemical composition of the bulk aerosol is still commonly applied in CCN studies (Zhang et al., 2014; Zhang et al., 2016; Che et al., 2017; Cai et al., 2018), especially when particle hygroscopicity measurements were in lack.*"

*15. L439-440, such a statement is not necessarily true. Primary particles can be CCN active. In addition, the authors defined kappa>0.1 as hygroscopic particle in the method part. Kappa of SOA can be <0.1, which contracts the statement here.*

Response: Thanks for your comments. We agree that POA can be CCN active and kappa of SOA can also be lower than 0.1. However, in general, SOA have higher hygroscopicity than POA (Frosch et al., 2011; Lambe et al., 2011; Kuang et al., 2020). The statement here has been revised as "*As SOA is generally considered to be more hygroscopic than POA (Frosch et al., 2011; Lambe et al., 2011; Kuang et al., 2020c), the increase of hygroscopic particles or SA particles (both SIA and SOA) were*

*considered to be the cause for the increase of SPAR within 200 to 300 nm size range (Fig. 2). In order to account for the variations of hygroscopic particles or SA particles in $N_{CCN}$ calculation, Number Fraction of hygroscopic particles (GF(90%, 200 nm)>1.22, $NF_{hygro}$) measured by HTDMA and Mass Fraction ...*"

16. *L454, "real-time MAF can be estimated by MF_SA", how to estimate, by simple linear regression?*

Response: The values of MF_SA were assumed to equal to MAF and used as real-time MAF to calculate SPAR and NCCN. We have revised this sentence as "*Thus, in the prediction of $N_{CCN}$, real-time SPAR can be calculated from average Da and MAF assumed to equal to real-time $MF_{SA}$ ($N_{CCN\_MF}$).*"

17. *L473, how do MAF and its diurnal variation depend on SS?*

Response: The diurnal variations of MAF at the five measured SSs are shown as follow:

[Figure]

**Fig. S5. Diurnal variations of MAF at the five measured SSs (indicated by different colors) during the high (left) and low (right) RH periods.**

As mentioned earlier, the diurnal variations of MAF at the five measured SSs reveal significant diurnal variations in MAF at low SSs (0.05% and 0.07%) that are dependent on RH conditions, while weaker diurnal variations that are insensitive on RH conditions at SSs over 0.2%. In general, MAF become lower at lower SSs, especially during nighttime. We have added this figure into the supplements and this discussion into the last paragraph of section 3,2 (the summary of this section) as "*The diurnal variations of MAF at the five measured SSs (Fig. S6) reveal significant diurnal variations in MAF at low SSs (0.05% and 0.07%) that are dependent on RH conditions, while small diurnal variations that are insensitive to the RH conditions at SSs over 0.2%. In general, MAF become lower at lower SSs, especially during nighttime.*".

*18. L495, in the abstract, 50% was used while here 40% was used...*

Response: Thanks for your comment and we have revised it accordingly.

*19. L509, "mixing state", is the right word here? What is the mixing state of these aerosols based on the "measurements of CCN activity, particle hygroscopicity and particle chemical compositions"?*

Response: Thanks for your comments. Here "mixing state" refers to MAF (SPAR parameter). To avoid confusion, we have revised it to "*MAF (SPAR parameter)*" in this sentence and also in the abstract accordingly.

*20. L797, in Fig. 2, it is helpful to describe the OA factors in the method part.*

Response: Thanks for your suggestion and we have added the description as follow:

"*including OA factors: hydrocarbon-like OA (HOA), cooking OA (COA), biomass burning OA (BBOA), coal combustion OA (CCOA), and oxygenated OA (OOA).*"

*Technical comments:*

*1. L214, "NF$_{hygro}$" was written as "NF_hygro" later.*

Response: Thanks for your suggestion. We have revised them accordingly.

*2. L272, "Dec." should be "Dec".*

Response: Thanks for your suggestion. We have revised it accordingly.

*3. L283, "are" should be "is". "Higher" might be better than "stronger".*

Response: Thanks for your suggestion. We have revised them accordingly.

*4. L324, "um" should be "μm".*

Response: Thanks for your suggestion. We have revised it accordingly.

*5. L346, "normalized" is missed before PM2.5? "Fig. 4(a1)" should be "Fig. 4(1a)".*

Response: Thanks for your suggestion. We have revised them accordingly.

*6. L356, "of" should be omitted.*

Response: Thanks for your suggestion. We have revised it accordingly.

*7. L376-378, this sentence is hard to understand.*

Response: Thanks for your comments. We have revised this sentence as "*SA formation mainly*

*enhanced number fraction of CCN-active particles in particle size of 200 to 300 nm, as SPAR only revealed evident enhancement (Fig. S2(b2)) and $N_{CCN}$ only significantly increased (Fig. 4(c2)) in that size range."*

*8. L432-433, "there were similar difference between CCN_AvgMAF and NCCN_meas" this sentence is hard to understand.*

Response: Thanks for your comments. We have revised this sentence as "*Only $N_{CCN\_AvgMAF}$ displayed similar deviations from $N_{CCN\_meas}$, indicating that differences between $N_{CCN\_cal}$ and $N_{CCN\_meas}$ were mainly contributed by variations in MAF brought on by significant CCN-active particles number fraction growth due to SA formations.*".

*9. L452, "the application of MF_SA on NCCN calculation were shown", it is in Fig. 6c rather than 6b.*

Response: Thanks for your suggestion. We have revised it accordingly.

*10. L467-468, this sentence is hard to understand.*

Response: Thanks for your comment. We have revised this sentence as "*Similar as before, $NF_{hygro}$ was applied as a proxy for MAF in the $N_{CCN}$ calculation, which also significantly improved the underestimation and correlation between $N_{CCN\_cal}$ and $N_{CCN\_meas}$ (Fig. 7(c)).*"

*11. L825, "the" should be "The".*

Response: Thanks for your suggestion. We have revised it accordingly.

Reference:

Rose, D., Gunthe, S. S., Mikhailov, E., Frank, G. P., Dusek, U., Andreae, M. O. and Pöschl, U.: Calibration and measurement uncertainties of a continuous-flow cloud condensation nuclei counter (DMT-CCNC): CCN activation of ammonium sulfate and sodium chloride aerosol particles in theory and experiment, Atmos. Chem. Phys., 8(5), 1153–1179, 2008.

Zhang, F., Li, Y., Li, Z., Sun, L., Li, R., Zhao, C., Wang, P., Sun, Y., Liu, X., Li, J., Li, P., Ren, G., and Fan, T.: Aerosol hygroscopicity and cloud condensation nuclei activity during the AC3Exp campaign: implications for cloud condensation nuclei parameterization, Atmos. Chem. Phys., 14, 13423–13437, https://doi.org/10.5194/acp-14-13423-2014, 2014.

Zhang, F., Li, Z., Li, Y., Sun, Y., Wang, Z., Li, P., Sun, L., Wang, P., Cribb, M., Zhao, C., Fan, T., Yang, X., and Wang, Q.: Impacts of organic aerosols and its oxidation level on CCN activity from measurement at a suburban site in China, Atmos. Chem. Phys., 16, 5413–5425, https://doi.org/10.5194/acp-16-5413-2016, 2016.

Che, H.C., Zhang, X.Y., Zhang, L. et al. Prediction of size-resolved number concentration of cloud condensation nuclei and long-term measurements of their activation characteristics. Sci Rep 7, 5819 (2017). https://doi.org/10.1038/s41598-017-05998-3

Cai, M., Tan, H., Chan, C. K., Qin, Y., Xu, H., Li, F., Schurman, M. I., Liu, L., and Zhao, J.: The size-resolved cloud condensation nuclei (CCN) activity and its prediction based on aerosol hygroscopicity and composition in the Pearl Delta River (PRD) region during wintertime 2014, Atmos. Chem. Phys., 18, 16419–16437, https://doi.org/10.5194/acp-18-16419-2018, 2018.

Hu, M., Peng, J., Sun, K., Yue, D., Guo, S., Wiedensohler, A., and Wu, Z.: Estimation of size-resolved ambient particle density based on the measurement of aerosol number, mass, and chemi- cal size distributions in the winter in Beijing, Environ. Sci. Tech- nol., 46, 9941–9947, https://doi.org/10.1021/es204073t, 2012.

Zhao, G., Tan, T., Zhao, W., Guo, S., Tian, P., and Zhao, C.: A new parameterization scheme for the real part of the ambient urban aerosol refractive index, Atmos. Chem. Phys., 19, 12875–12885, https://doi.org/10.5194/acp-19-12875-2019, 2019.

Frosch M, Bilde M, DeCarlo PF, Jurányi Z, Tritscher T, Dommen J, et al. Relating cloud condensation nuclei activity and oxidation level of α-pinene secondary organic aerosols. J Geophys Res- Atmos. 2011;116(D22). https://doi.org/10.1029/2011jd016401.

Lambe AT, Onasch TB, Massoli P, Croasdale DR, Wright JP, Ahern AT, et al. Laboratory studies of the chemical composition and cloud condensation nuclei (CCN) activity of secondary organ- ic aerosol (SOA) and oxidized primary organic aerosol (OPOA). Atmos Chem Phys. 2011;11(17):8913–28. https://doi.org/10. 5194/acp-11-8913-2011.

Kuang, Y., Xu, W., Tao, J. et al. A Review on Laboratory Studies and Field Measurements of

Atmospheric Organic Aerosol Hygroscopicity and Its Parameterization Based on Oxidation Levels. Curr Pollution Rep 6, 410–424 (2020). https://doi.org/10.1007/s40726-020-00164-2

---

## Referee Report (RR1)

The manuscript has improved a lot since the first version, therefore now I can imagine it being accepted for publication. However, I still have some comments and recommendations for the authors. And I still have to say, another language editing for the second half of the paper would be really nice. It is still very hard to read.

Line 239: black carbon is assumed to be hydrophilic -> non-hydrophilic or non-hygroscopic

Line 338-339: GF-PDF, SPAR is a function of the diameter, if you discuss an increase (or whatever) of these properties, please always indicate at which diameter you mean it

Line 343: was be found: please remove "be"

Line 361-386: Please discuss the fit parameters, the main parameters of the SPAR curves rather than the SPAR value at a certain diameter (at 300nm). Looking at Figure 3 (b and c) makes it clear what happens, and I do not read that from the paper, as it is written now. What I see there is, that Da values are almost exactly the same at low and high RH, which means that the hygroscopicity of this particle population is the same at high and low RH, during the SA formation there is for both cases a little decrease in Da which means a slight hygroscopicity increase due to the more hygroscopic SA material. There is a significant difference between the MAF values, but outside of the SA formation, so the fraction of the (as you call it) hygroscopic particles at low RH was lower in PA, and good part of this difference disappears during the presence of the SA formation. And with this the message for me would be, the hygroscopicity of the SA formed under low and high RH conditions is very similar, and the difference what you see in N_CCN is mainly because the mass/number and size of formed SA particles is different at low and at high RH.

Line 419-422: Please delete/correct these sentences, they are not correct. You cannot compare volume to number like this. You should compare N_CCN to the number concentration, that makes sense. But you have already discussed what happens with the SPAR value, which is exactly the same, the ratio of them.

Line 431 and 342: within -> below

Line 441-451: Figure 3b shows that the hygroscopicity increase during SA formation was very similar in both low and high RH cases

Line 486: variation of MAF = changes in aerosol hygroscopicity is not true. You write that yourself one sentence before that the MAF represents the fraction of the hygroscopic particles. Please change this sentence.

Line 487-494: Please let the reader know here, that this, what you talk about (trying to estimate the changes of the MAF from other measurements) here will be shown in the next part of the paper

Line 495-497: please rephrase this sentence, very hard to understand

Line 503: can be -> is

Figures 6 and 7: taking into account that MAF is less than 1 makes your prediction much better. What is not clearly stated, where is the Da coming from in this case? Please indicate it. It should be the chemistry/growth factor derived Da and not the one which was originally derived from the SPAR curves.

---

## Author Response (AR2)

Dear Editor,

We greatly thank the reviewers for their review. Point-by-point responses addressing the reviewers' comments were uploaded (also attached to this file). The manuscript has been revised and improved accordingly.

Yours sincerely,

Jiangchuan Tao and Nan Ma

**Response to reviewers' comments on manuscript (acp-2020-939)**

(Reviewer comments in italics, the responses in plain font)

**Reviewer #1**

The manuscript has improved a lot since the first version, therefore now I can imagine it being accepted for publication. However, I still have some comments and recommendations for the authors. And I still have to say, another language editing for the second half of the paper would be really nice. It is still very hard to read.

Response: Thanks for your comments. Suggestions and comments are addressed point-by-point and corresponding responses are listed below. In addition, we have improved the language in the second half of the paper.

Specific Comments:

*Line 239: black carbon is assumed to be hydrophilic -> non-hydrophilic or non-hygroscopic* Response: Thanks for your suggestion. We have replaced "hydrophilic" with "non-hygroscopic".

Line 338-339: GF-PDF, SPAR is a function of the diameter, if you discuss an increase (or whatever) of these properties, please always indicate at which diameter you mean it

Response: Thanks for your comments. Here we are referring to increase of SPAR in particle size range from 200nm to 400nm and increase of GF-PDF in GF range from 1.2 to 1.8. We have revised this sentence as: "Simultaneous daytime increases in particle SPAR in particle size range from 200 nm to 400 nm, GF-PDF in GF range from 1.2 to 1.8 and SA mass fraction were found in all four events, suggesting that SA formation led to increasing hygroscopic particles number concentration, which in turn enhanced particle CCN activity."

Line 343: was be found: please remove "be"

Response: Thanks for your suggestion. We have revised it accordingly.

Line 361-386: Please discuss the fit parameters, the main parameters of the SPAR curves rather than the SPAR value at a certain diameter (at 300nm). Looking at Figure 3 (b and c) makes it clear what happens, and I do not read that from the paper, as it is written now. What I see there is, that Da values are almost exactly the same at low and high RH, which means that the hygroscopicity of this particle population is the same at high and low RH, during the SA formation there is for both cases a little decrease in Da which means a slight hygroscopicity increase due to the more hygroscopic SA material. There is a significant difference between the MAF values, but outside of the SA formation, so the fraction of the (as you call it) hygroscopic particles at low RH was lower in PA, and good part of this difference disappears during the presence of the SA formation. And with this the message for me would be, the hygroscopicity of the SA formed under low and high RH conditions is very similar, and the difference what you see in N\_CCN is mainly because the mass/number and size of formed SA particles is different at low and at high RH.

Response: Thanks for your suggestion. We have added the corresponding discussions in the end of this paragraph as follow:

"In detail, the different variations of SPAR in high and low RH events indicated by MAF and  $D_a$  shown in Figs. 3(b & c) suggested different variations of hygroscopicity, number fraction and size of SA particles. Before SA formation, there was a significant difference between the MAF in high and low RH events, which disappeared after the SA formation. The stronger variations in MAF in low RH events suggested stronger enhancement of number concentration of formed SA particles. As for  $D_a$ during SA formation, there were similar, little decrease in both high and low RH events, suggesting similar hygroscopicity of the SA formed under low and high RH conditions. Thus differences of SPAR and the resultant  $N_{CCN}$  during low and high RH events were mainly due to the different variations of number fraction of formed SA particles."

Line 419-422: Please delete/correct these sentences, they are not correct. You cannot compare volume to number like this. You should compare  $N_{CCN}$  to the number concentration, that makes sense. But you have already discussed what happens with the SPAR value, which is exactly the same, the ratio of them.

Response: Thanks for your comments. These sentences are incorrect and have been deleted.

**Line 431 and 342: within -> below**

Response: Thanks for your suggestion. We have revised them accordingly.

**Line 441-451: Figure 3b shows that the hygroscopicity increase during SA formation was very similar in both low and high RH cases**

Response: Thanks for your comments. It should be the increase of number fraction of SA rather than SA hygroscopicity which leads to higher NCCN enhancement. We have revised the corresponding discussions in the manuscript as:

"Under high RH conditions, there were strong SIA dominated SA formation leading to stronger enhancements of CCN-active particle number fraction and  $N_{CCN}$ . Meanwhile, under low RH conditions,

there were moderate SOA dominated SA formation with moderate enhancements of CCN-active particle number fraction and  $N_{CCN}$ ."

Line 486: variation of MAF = changes in aerosol hygroscopicity is not true. You write that yourself one sentence before that the MAF represents the fraction of the hygroscopic particles. Please change this sentence.

Response: Thanks for your suggestion. It should be the fraction of the hygroscopic particles and we have revised this sentence as "*Thus, for accurate*  $N_{CCN}$  estimations, considering the variation of MAF (changes in fraction of the hygroscopic particles) is highly essential."

Line 487-494: Please let the reader know here, that this, what you talk about (trying to estimate the changes of the MAF from other measurements) here will be shown in the next part of the paper

Response: Thanks for your suggestion. We have revised the last sentence in this paragraph by adding the corresponding description as: "In order to account for the variations of hygroscopic particles or SA particles in  $N_{CCN}$  calculation, in the following part, Number Fraction of hygroscopic particles (GF(90%, 200 nm)>1.22, NFhygro) measured by HTDMA and Mass Fraction of SA particles (MFSA) measured by ACSM in this campaign were used to represent MAF variations and to provide calculation of  $N_{CCN}$  at SS of 0.05% …"

**Line 495-497: please rephrase this sentence, very hard to understand**

Response: Thanks for your suggestion. We have revised it as: "Based on the bulk hygroscopicity derived from particle chemical compositions measurements ( $\kappa_{chem}$ ), a critical diameter for CCN activation can be calculated based on  $\kappa$ -Köhler theory. With this critical diameter,  $N_{CCN}(0.05\%)$  can be predicted incorporating measured PNSD ( $N_{CCN Chem}$ )."

**Line 503: can be -> is**

Response: Thanks for your suggestion. We have revised it accordingly.

Figures 6 and 7: taking into account that MAF is less than 1 makes your prediction much better. What is not clearly stated, where is the Da coming from in this case? Please indicate it. It should be the chemistry/growth factor derived Da and not the one which was originally derived from the SPAR curves.

Response: Thanks for your comments.

In figures 6 and 7, we evaluate the calculation of  $N_{CCN}$  by using  $NF_{hygro}$  or  $MF_{SA}$  as the SPAR

parameter MAF, which was found to contribute the most of the calculated  $N_{CCN}$  deviation in Figure 5, especially during low RH periods. Besides neglecting the MAF variations, the differences between MAF and  $NF_{hygro}$  (or MFSA, shown in figure 6b or 7b) and the application of campaign average Da (Figure 5) can both contribute to the deviation of  $N_{CCN}$  calculation. Thus, in order to highlight the application of using  $NF_{hygro}$  or MFSA as estimation of MAF variations, the campaign average Da of SPAR curves in Figure 5 was used in the calculation of SPAR and  $N_{CCN}$  in figures 6 and 7. As the deviation of  $N_{CCN}$  calculation by using  $NF_{hygro}$  or MFSA as MAF became smaller than those without considering MAF variations, it was concluded that  $NF_{hygro}$  and MFSA can be used as MAF to improve  $N_{CCN}$  prediction.

Figure R1. (a and b)The comparison between measured NCCN and calculated NCCN based on SPAR derived from real-time MFSA and GF derived Da (NCCN\_HTDMA, a) or chemistry derived Da
(NCCN\_ACSM, b). The black dashed lines represent the relative deviation of 30%. (c) Time series of GF derived Da (blue markers), chemistry derived Da (yellow markers) and the campaign averaged Da from SPAR curves (green line).

In addition, we also agree that it's important to apply chemistry/GF derived Da in  $N_{CCN}$  calculation as the reviewer suggested. However, as there can be significant deviations between CCN activity measured under different water vapor saturated conditions, there may be larger  $N_{CCN}$  deviations by using real-time chemistry/growth factor derived Da rather than a campaign average Da, which can be obtained from other CCN measurement in the same areas. In the figure R1 shown above,  $N_{CCN}$ calculated with chemistry/growth factor derived Da were compared with the measured  $N_{CCN}$ . As shown in figure R1(a), by using Da calculated from GF-PDF, the deviations of calculated  $N_{CCN}$  ( $N_{CCN_HTDMA}$ ) become smaller, although there can be underestimation larger than 30%. This underestimation by using Da calculated from GF-PDF may be due to underestimating hygroscopicity of hygroscopic particles measured by HTDMA under sub-saturated conditions (higher values of yellow markers than the blue line in figure R1(c)). In Figure S3, similar underestimations (larger than 30%) of  $N_{CCN}$  calculated with GF-PDF can also be found. In Figure R1 (b), the deviations of calculated  $N_{CCN}$  chemistry derived Da ( $N_{CCN_ACSM}$ ) become smaller, although there can be overestimation larger than 30%. These smaller deviations may be due to better agreement between Da of SA (green markers in figure R1(c)) and campaign averaged  $D_a$  from SPAR curves. However, this better agreement of Da may result from the combination of overestimated CCN hygroscopicity by neglecting CCN-active PA and underestimation of SOA hygroscopicity under super-saturated conditions. Furthermore, the difference of CCN hygroscopicity measured by different instruments which contributed to these  $N_{CCN}$  deviations, was not the focus of this study. Thus, the chemistry/growth factor derived Da were not applied on  $N_{CCN}$  calculation in this study.

We have made the descriptions of Da clear in the manuscript as follow:

Line 502-504, in the end of paragraph before figure 6 and 7: "To be noted, in order to highlight the application of using  $MF_{SA}$  as estimation of MAF variations on  $N_{CCN}$  calculation, the campaign averaged  $D_a$  from SPAR curves was used."

Line 520-522, the paragraph for figure 6: "Thus, in the prediction of  $N_{CCN}$ , real-time SPAR can be calculated from  $D_a$  and MAF assumed to equal to real-time  $MF_{SA}$  ( $N_{CCN_MF}$ )."

Line 538-539, the paragraph for figure 7: "Also, the campaign averaged  $D_a$  in Fig. 5a.was used to calculate SPAR curves and  $N_{CCN}$ ."

**Reviewer #2:**

The responses and the revised manuscript have largely addressed my previous comments. However, a few minor problems still remain.

Response: Thanks for your comment. Suggestions and comments are addressed point-by-point and corresponding responses are listed below.

1. Regarding my former general comment #1, I suggest the authors to explicitly specify supersaturation when describing NCCN in order to avoid ambiguity and over-generalization of the findings. For example, in the abstract (L36 and L570) and conclusion, NCCN can be changed to "NCCN(0.05%)".

Response: Thanks for your suggestion. We have revised them accordingly and also made corresponding revisions throughout the manuscript.

**2. In the responses, Pg 34, the sentence of L4-6 were repeated.**

Response: Thanks for your comment. These sentences were incorrectly repeated in the responses, but the corresponding discussions in the manuscript have been confirmed to be correct.

3. Regarding my previous specific comment #2, "L264, how are PA and SA characterized?" I meant in which method PA and SA were determined/quantified.

Response: Thanks for your comment. PA (including four POA) and SA (including OOA and inorganic compounds) were determined based on ACSM measurement. In detail, OOA and four POA were analyzed based on the ACSM PMF analysis (Zhang et al. 2011). We have revised the sentence as "*The timeseries of meteorological parameters, SPAR, NCCN at SS of 0.05% and mass concentration of Non-refractory particulate matter of PM2.5 (NR-PM2.5), PM2.5 SA (inorganic compounds and OOA) and PM2.5 PA (primary aerosol, defined as the sum of POA) are shown in Fig. 1. The mass concentration of OOA and four POA were quantified by the ACSM PMF analysis (Zhang et al. 2011)."*

**Reference:**

Zhang, Q., Jimenez, J. L., Canagaratna, M. R., Ulbrich, I. M., Ng, N. L., Worsnop, D. R., and Sun, Y.: Understanding atmospheric organic aerosols via factor analysis of aerosol mass spectrometry: a review, Analytical and Bioanalytical Chemistry, 401, 3045–3067, https://doi.org/10.1007/s00216-011-5355-y, 2011.

---

## Author Response (AR3)

Dear Editor,

Thank you very much for your attention and consideration. The data availability part are modified as:

**"Data availability.**

The data used in this study are available from the file sharing link (https://pan.baidu.com/s/1iSMdEQj_KRrjmmrXtXNJUg) using extracting code db2p."

Yours sincerely,

Jiangchuan Tao and Nan Ma